# A Tight Error Bound for Deep Learning via Distribution and Loss Complexity

## Abstract

Generalization is a central requirement for machine learning models in real-world applications, yet theoretically verifying it for trained models - especially deep neural networks (NNs) - remains highly challenging. Existing generalization bounds are often vacuous for modern NN architectures. In this paper, we propose model-dependent bounds that connect a model's behavior in data space with its generalization ability. Our bounds explicitly capture both the complexity of the data distribution and the loss function, and they highlight the role of alignment between data geometry and the loss landscape. These properties enable our bounds to obtain significantly tighter estimates of test error than prior approaches. Extensive experiments on ImageNet classification and segmentation models show that our tractable bound consistently provides the tightest (nonvacuous) test-error estimates across a wide range of large-scale NNs. Moreover, we find that some parts of our bounds can effectly track the dynamic of test error, offering new insights into how to understand and improve performance in deep learning.

## 1 Introduction

A central challenge in modern machine learning is to understand and rigorously characterize the generalization ability of deep neural networks (NNs). Despite their massive parameter counts and ability to interpolate complex training data (Zhang et al., 2021), deep NNs often generalize remarkably well to unseen data. This apparent paradox has motivated a substantial body of theoretical research, aiming to explain why overparameterized models do not necessarily overfit, and to develop principled error guarantees that can serve as a foundation for deep learning theory.

Early efforts connect weight norms to Rademacher complexity and show that smaller norms imply tighter error bounds (Bartlett et al., 2017; Golowich et al., 2020; Neyshabur et al., 2015; 2017). However, practical NNs often have large spectral or Frobenius norms, rendering these bounds vacuous in practice (Arora et al., 2018). Alternative approaches based on algorithmic stability (Bousquet & Elisseeff, 2002; Shalev-Shwartz et al., 2010) and robustness (Xu & Mannor, 2012; Kawaguchi et al., 2022) provide intuitive connections between sensitivity and generalization, yet require overly strong assumptions (such as stability under sample replacement or local smoothness of the loss landscape) often violated in modern architectures which are prone to stochasticity and adversarial vulnerabilities (Madry et al., 2018; Goodfellow et al., 2015). Mutual information (MI) frameworks (Xu & Raginsky, 2017; Tishby & Zaslavsky, 2015) further highlight the trade-off between compression and predictive accuracy, but their applicability is hindered by the intractability of computing MI in high-dimensional nonlinear regimes.

More recently, PAC-Bayes theory has emerged as a powerful tool, yielding the first non-vacuous bounds for certain stochastic networks (Dziugaite & Roy, 2017; Zhou et al., 2019; Lotfi et al., 2022). Progress has extended these ideas to larger architectures, adversarial robustness, and even large language models (Lotfi et al., 2024a;b), but often at the cost of compression, quantization, or posterior sampling. A significant change (e.g. compression or quantization) can cause those bounds not to provide guarantees to the original trained model of interest. Efforts toward deterministic PAC-Bayes bounds (Biggs & Guedj, 2022; Viallard et al., 2024; Clerico et al., 2025) provide valuable insights, yet so far yield meaningful guarantees only for shallow or simplified architectures. A promising step forward is the recent work of Than & Phan (2025), who established non-vacuous bounds for large-scale NNs without requiring model modification. However, their computable guarantees re-

main loose, as they exploit little information about the loss landscape (mostly through the training loss) and scale with $\sqrt{\ln K}$, where $K$ is the size of the partition of the data space.

Our contributions are threefold:

- We introduce a novel bound for the test error of a given (trained) model. This bound reveals the crucial role of the complexity of the data distribution and the loss function (by capturing both micro-level and macro-level behaviors of the model) to the test error. It further uncovers a novel interaction between data geometry and the loss landscape, which has not been characterized in prior theory.
- We derive an exactly computable bound that retains the structural form of our general bound while being fully estimable from the training set under only i.i.d. and bounded-loss assumptions. Unlike previous bounds, our bound avoids $\sqrt{\ln K}$ scaling and thus can provide substantially sharper estimates of test error.
- As a key technical step to develop our error bounds, we establish a new concentration inequality for multinomial random variables. This result eliminates the $\sqrt{\ln K}$ dependency that appears in prior analyses, yielding arguably tighter convergence rates for high-dimensional settings.

We validate our exactly computable bound through extensive experiments on 32 large-scale NN classifiers pretrained by PyTorch on ImageNet and 5 semantic segmentation models pretrained on COCO. Across these 37 networks, our bound consistently yields significantly tighter estimates of test error than prior bounds. Moreover, we observe strong empirical correlation between the macro-level behavior and the true test errors. *Surprisingly, while validation error sometimes cannot track the test error, the macro-level behavior in our bound can.* Those facts suggest that the proposed framework can serve as a reliable diagnostic tool for model evaluation and selection. These results demonstrate that our bound provide meaningful theoretical guarantees for trained models at deployment scale.

## 2 RELATED WORK

*Theories for generalization ability:* Early theoretical efforts to explain the generalization of NNs focused on bounding complexity through norms and capacity measures. Norm-based approaches (Bartlett et al., 2017; Golowich et al., 2020; Neyshabur et al., 2015; 2017; Galanti et al., 2023b) connected weight norms to Rademacher complexity, showing that smaller norms lead to tighter generalization guarantees. However, as pointed out by Arora et al. (2018), practical NNs often have large spectral or Frobenius norms, making the resulting bounds vacuous. Relatedly, algorithmic stability (Bousquet & Elisseeff, 2002; Shalev-Shwartz et al., 2010; Charles & Papailiopoulos, 2018; Kuzborskij & Lampert, 2018) and robustness (Xu & Mannor, 2012; Sokolić et al., 2017; Kawaguchi et al., 2022; Than et al., 2025) provide alternative perspectives by linking generalization to the sensitivity of algorithms with respect to perturbations in the data. Stable algorithms, whose outputs are unaffected by small training set changes, are provably generalizable, but impractical, since two different runs of popular (stochastic) algorithms often produce slightly different models. Similarly, robustness-based bounds assume local smoothness of the loss landscape around training points, yet in practice, modern NNs are highly sensitive and prone to adversarial examples (Szegedy et al., 2014; Goodfellow et al., 2015; Madry et al., 2018; Zhou et al., 2022). Mutual information (MI)-based approaches (Xu & Raginsky, 2017; Wang et al., 2021; Sefidgaran et al., 2022; Nadjahi et al., 2024) offer a principled framework by quantifying dependencies between data and representations. While MI theories capture the trade-off between compression and accuracy (Tishby & Zaslavsky, 2015), their practical use in deep learning is constrained by the intractability of computing MI in high-dimensional, nonlinear regimes.

*Recent progress:* A major breakthrough comes from PAC-Bayes theory (McAllester, 1999; Neyshabur et al., 2018; Haddouche & Guedj, 2023; Biggs & Guedj, 2023), which provides data-dependent, non-vacuous generalization bounds under certain conditions. By optimizing posterior distributions over neural network weights, Dziugaite & Roy (2017) demonstrated the first non-vacuous PAC-Bayes bounds for small stochastic NNs. Building on this, Zhou et al. (2019) and Lotfi et al. (2022) used compression-based arguments to derive non-vacuous bounds for larger networks such as LeNet and MobileNet trained on ImageNet. Recent advances extend these ideas to adversarial robustness (Mustafa et al., 2024), prompts (Akinwande et al., 2024), and even convolutional networks up to 20 layers (Galanti et al., 2023a). Strikingly, Lotfi et al. (2024a;b) pushed the frontier

by providing the first non-vacuous bounds for large language models, through PAC-Bayes combined with quantization, finetuning, and token-level analysis. Nevertheless, *significant limitations remain: these bounds often apply only to quantized or compressed models, rather than their original trained versions*. Efforts toward deterministic PAC-Bayes bounds (Viallard et al., 2024; Clerico et al., 2025; Biggs & Guedj, 2022) suggest promising directions, but so far they yield non-vacuous guarantees only for shallow or simplified architectures. Thus, while progress is being made, the overarching challenge of producing tight, non-vacuous, and practically computable bounds for large-scale NNs remains largely unresolved.

Than & Phan (2025) have recently made important progress by deriving non-vacuous generalization bounds for large-scale NNs without requiring modifications to the models or imposing strong assumptions. Their results demonstrate that such bounds can capture the complexity of the data distribution and partially leverage local behaviors of the model, thereby highlighting a promising direction to develop practically useful error guarantees. However, their approach has two notable limitations: (1) the exactly computable bound relies only on the training error and cannot fully exploit richer local behaviors of the model, which makes the guarantee relatively loose; and (2) their bounds scale with $\sqrt{\ln K}$, where $K$ is the partition size, which is suboptimal. In contrast, our bounds in this work is able to incorporate both micro- and macro-level behaviors of the model while explicitly encoding the complexity of the data distribution. More importantly, our bounds does not directly scale with $\sqrt{\ln K}$, owing to our novel concentration bound for multinomial random variables. Together, these properties make our bounds more favorable and provide significantly tighter estimates of the test error.

## 3 ERROR BOUNDS

In this section, we present two novel bounds for the test error of a given (possibly trained) model. The first bound provides a general form which directly depends on micro and macro behaviors of the trained model at different small areas of the data space. The second bound is easily computed from the training set.

### 3.1 PRELIMINARIES

We will work with a model $h$, defined on a data space $\mathcal{Z}$, and a nonnegative *measuring (loss) function* $\ell$ to measure $\ell(h, z)$ which tells the quality of the prediction of model $h$ about a given instance $z$. For a given model $h$, we often want to know *expected loss* $F(P, h) = \mathbb{E}_{z \sim P}[\ell(h, z)]$, which tells the overall quality of $h$ w.r.t. the data distribution $P$. However, this quantity cannot be computed exactly and sometimes is approximated by the *empirical loss* $F(S, h) = \frac{1}{n} \sum_{z \in S} \ell(h, z)$ computed on a data set $S = \{z_1, ..., z_n\} \subseteq \mathcal{Z}$. When using the empirical loss, we often miss some uncertainty about model $h$.

*Notations:* $|S|$ denotes the cardinality/size of a set $S$, $[K]$ denotes the set $\{1, ..., K\}$ for a given natural number $K$. Let $\Gamma(\mathcal{Z}) := \bigcup_{i=1}^{K} \mathcal{Z}_i$ be a partition of $\mathcal{Z}$ into $K$ disjoint nonempty subsets. Denote $S_i = S \cap \mathcal{Z}_i$, $n_i = |S_i|$, $n = \sum_{j=1}^{K} n_j$, and $T = \{i \in [K] : n_i > 0\}$.

### 3.2 MAIN RESULTS

We consider how well a trained model can generalize on unseen data. The following result is proven in Appendix A.

**Theorem 3.1** (General bound). *Given a partition $\Gamma$ and a bounded non-negative loss $\ell$, consider a model $h$ which may depend on a dataset $S$ with $n$ i.i.d. samples from distribution $P$. For any constants $\gamma \geq 1, \delta_1 > 0, \delta_2 \geq \exp(-\frac{u \ln \gamma}{\max\{4b, 8n-6\}})$, the following holds with probability at least $1 - \delta_1 - \delta_2$ over the sampling of $S$:*

$$F(P, h) \leq F(S, h) + \frac{b}{n} \sum_{i \in T} F(S_i, h) + C\sqrt{\frac{u \ln (1/\delta_2)}{2n^2}} + a_o\sqrt{\frac{\ln(1/\delta_1)}{2n}} \tag{1}$$

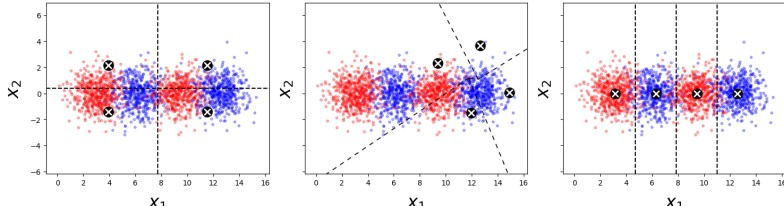

Figure 1: Examples about the data geometry and partition. The dots are the samples generated from a mixture model with 4 components, each color represents a class, while the "x" dots are the centroids that build a partition. The left subfigure shows a partition $\Gamma$ that divides the data space into areas of comparable size, the middle figure shows a partition whose centroids are generated randomly, while the right-most subfigure shows a partition with balanced local probabilities.

*where $b = \sqrt{0.5n \ln(1/\delta_1)}$, $u = \gamma n(1 + 2b) + |\boldsymbol{T}|b^2 + \gamma^2 n^2 (\sum_{i=1}^{K} p_i^2)$, $p_i = P(\mathcal{Z}_i)$ is the measure of area $\mathcal{Z}_i$ for $i \in [K]$, $C = \sup_{\boldsymbol{z} \in \mathcal{Z}} \ell(\boldsymbol{h}, \boldsymbol{z})$, $a_i(\boldsymbol{h}) = \mathbb{E}_{\boldsymbol{z} \sim P}[\ell(\boldsymbol{h}, \boldsymbol{z}) | \boldsymbol{z} \in \mathcal{Z}_i]$ for $i \in [K]$, and*

$$a_o = \begin{cases} 0 & \textit{if } |\boldsymbol{T}| = K, \\ (\sum_{k \notin \boldsymbol{T}} p_k)^{-1} \sum_{k \notin \boldsymbol{T}} p_k a_k(\boldsymbol{h}) & \textit{otherwise} \end{cases}$$

This theorem establishes that the test error of a model $\boldsymbol{h}$ can be upper bounded by the right-hand side of (1), which consists of the empirical error on $\boldsymbol{S}$ and an additional uncertainty term. Importantly, the bound holds for any specific model $\boldsymbol{h}$, regardless of whether or not it has been trained on $\boldsymbol{S}$. A closer examination reveals several interesting properties of bound (1):

- *Encoding the complexity of the loss:* Bound (1) highlights that **both the micro- and macro-level behaviors of $\boldsymbol{h}$ play a central role in generalization**. While the empirical loss $F(\boldsymbol{S}, \boldsymbol{h})$ captures the average prediction accuracy of $\boldsymbol{h}$ over individual samples, the quantity $mac_h = \sum_{i \in \boldsymbol{T}} F(\boldsymbol{S}_i, \boldsymbol{h})$ summarizes the prediction quality across local regions of the data space. Hence, the bound suggests that strong generalization requires $\boldsymbol{h}$ to perform well not only on individual predictions but also consistently across all local regions – an aspect absent in prior bounds.

- *Encoding the complexity of the data distribution:* The term $\sum_{i=1}^{K} p_i^2$ encodes the squared norm of $(p_1, \ldots, p_K)$, thus reflecting the shape and complexity of the distribution $P$ within local regions. For a fixed partition, structured distributions (e.g., Gaussian) can yield higher values of $\sum_{i=1}^{K} p_i^2$, making $u$ a direct measure of data complexity. Unlike most prior bounds, our result incorporates this distributional structure explicitly, a feature shared only with (Than & Phan, 2025).

- *The central role of the alignment between the data geometry and partition:* A well-chosen partition $\Gamma$ (e.g. the one on the right-most subfigure of Figure 1) that aligns with the geometry of $P$ can produce balanced local probabilities ($p_i \approx p_j$), thereby reducing $\sum_{i=1}^{K} p_i^2$ and $u$. In contrast, a poor alignment (e.g. the one on the middle of Figure 1) inflates $u$. This underscores the importance of data-partition alignment for achieving tight bounds, though selecting such partitions is challenging when $P$ is often unknown in practice.

- *Exhibiting a close interaction between the complexity of the data distribution and the loss, via the partition:* One way to make $\sum_{i=1}^{K} p_i^2$ smaller is to increase $K$, i.e., choose a finer-grained partition. For example, if one can partition the data space into areas with $K = O(n^{0.5})$ balanced probabilities (i.e., $p_i \approx p_j$ for all $i, j$), then our bound (1) will be $F(\boldsymbol{S}, \boldsymbol{h}) + \frac{b}{n} mac_h + O(n^{-0.25})$. Nonetheless, an increase of $K$ may lead to an increase in $|\boldsymbol{T}|$, suggesting an increase in the macro loss $mac_h$. On the other hand, a decrease in $K$ can lead to smaller $mac_h$, but potentially increase $\sum_{i=1}^{K} p_i^2$ and $u$. This trade-off **reveals a novel, explicit interaction between data distribution complexity and the loss landscape of $\boldsymbol{h}$**, which has not been captured in prior theory.

- *Model-dependence with mild assumptions:* The bound directly estimates the test error of $\boldsymbol{h}$ under only i.i.d. and bounded-loss assumptions. This practicality contrasts with most prior bounds that rely on stronger conditions.

*Proof sketch.* We first show

$$\Pr\left(F(P, \boldsymbol{h}) \geq \frac{1}{n}\sum_{i \in \boldsymbol{T}}(n_i + b)a_i(\boldsymbol{h}) + a_o\sqrt{\frac{\ln(1/\delta_1)}{2n}}\right) \leq \delta_1 \tag{2}$$

This is a consequence of Theorem 4.1, which provides a novel data-dependent concentration bound for multinomial random variables. The remaining task is to show the following:

$$\Pr\left(\frac{1}{n}\sum_{i \in \boldsymbol{T}}(n_i + b)a_i(\boldsymbol{h}) \geq \frac{1}{n}\sum_{i \in \boldsymbol{T}}(n_i + b)F(\boldsymbol{S}_i, \boldsymbol{h}) + C\sqrt{\frac{u}{2n^2}\ln\frac{1}{\delta_2}}\right) \leq \delta_2 \tag{3}$$

Combining this with (2) and the union bound will complete the proof.

We resolve the challenge (3) by developing Theorem A.1. Its proof contains three main steps:

1. First, denoting $B_K = \sum_{i=1}^{K}(n_i + b_i)a_i(\boldsymbol{h}) - \sum_{i=1}^{K}(n_i + b_i)F(\boldsymbol{S}_i, \boldsymbol{h})$ and $\boldsymbol{n} = \{n_1, ..., n_K\}$, we show the following for some constants $t$ and $y$: $\Pr(B_K \geq t) \leq e^{-yt}\mathbb{E}_{\boldsymbol{h},\boldsymbol{n}}\left[\mathbb{E}_{\boldsymbol{S}}\left[e^{yB_K}|\boldsymbol{h}, \boldsymbol{n}\right]\right]$

2. We next estimate $\mathbb{E}_{\boldsymbol{S}}\left[e^{yB_K}|\boldsymbol{h}, \boldsymbol{n}\right]$. Overall, we make sure that $\mathbb{E}_{\boldsymbol{S}}\left[e^{yB_K}|\boldsymbol{h}, \boldsymbol{n}\right] \leq e^{\psi(y, \boldsymbol{n})}$, for some function $\psi(y, \boldsymbol{n})$ which does not depend on $\boldsymbol{h}$. As a result $\Pr(B_K \geq t) \leq \mathbb{E}_{\boldsymbol{n}}e^{\psi(y, \boldsymbol{n})}$.

3. The last step is to bound $\mathbb{E}_{\boldsymbol{n}}e^{\psi(y, \boldsymbol{n})}$. This requires us to develop various analyses for binomial variables. A suitable choice for $t, y$ completes our proof in Appendix A. □

Though providing various insights into a model, our bound (1) cannot be exactly computed from a finite sample. The main reason comes from the unknown quantities $p_k$ and $a_k$. Therefore, the next theorem provides another bound which can be computed easily in practice.

**Theorem 3.2** (Computable bounds). *Given a partition $\Gamma$ and a bounded non-negative loss $\ell$, consider a model $\boldsymbol{h}$ which may depend on a dataset $\boldsymbol{S}$ with $n$ i.i.d. samples from distribution $P$. For any constants $\gamma \geq 1, \delta > 0$ and $\alpha \in [0, \frac{\gamma n(1+2b)+|\boldsymbol{T}|b^2+\gamma^2 n^2/K}{\max\{4b, 8n-6\}}]$, with probability at least $1 - \gamma^{-\alpha} - \delta$:*

$$F(P, \boldsymbol{h}) \leq F(\boldsymbol{S}, \boldsymbol{h}) + \frac{b}{n}\sum_{i \in \boldsymbol{T}}F(\boldsymbol{S}_i, \boldsymbol{h}) + C\sqrt{\hat{u}\alpha\ln\gamma} + C\sqrt{\frac{\ln(2/\delta)}{2n}} \tag{4}$$

*where $b = \sqrt{0.5n\ln(2/\delta)}$, $\hat{u} = \frac{\gamma(1+2b)}{2n} + \frac{\gamma|\boldsymbol{T}|b^2}{2n^2} + \frac{\gamma^2}{2}\sum_{i=1}^{K}\left(\frac{n_i}{n}\right)^2 + \gamma^2\sqrt{\frac{\ln(2/\delta)}{2n}}$.*

From its proof in Appendix A, one can observe that bound (4) is worse than bound (1). Luckily, it has two main advantages: *it can maintain the important properties of bound (1)*, and *it can be computed exactly from the training set, without missing any uncertainty*.

**Comparison with related model-dependent bounds:**

- *Non-computable bounds.* Most prior non-computable generalization bounds, including those based on norms (Bartlett et al., 2017; Galanti et al., 2023b), stability (Li et al., 2024), robustness (Kawaguchi et al., 2022), and mutual information (Nadjahi et al., 2024), provide valuable theoretical insights but are ultimately limited in practice because they either rely on strong assumptions (e.g., stability, robustness) or are vacuous (Than et al., 2025) or are intractable to compute from the training set only. PAC-Bayes theory (Lotfi et al., 2022; 2024b) has made major progress by establishing non-vacuous bounds, yet these often require strong modifications (e.g., quantization, compression) that depart from the original trained models.

- *Exactly computable bounds.* Than & Phan (2025) delivered an exactly computable guarantee which is non-vacuous for large-scale NNs. Despite being practically attractive, their guarantee tends to be loose. Indeed, their bound scales with $\sqrt{\ln K}$, which becomes suboptimal for large $K$. Our computable bound (4) achieves a better balance. Although being weaker than the general bound (1), it retains the key structural insights, capturing both micro- and macro-level behaviors and explicitly encoding distribution complexity while remaining exactly computable from the training set. While the training loss is the main way to exploit the loss landscape in the bounds by Than & Phan (2025), our bound (4) can

deeply exploit the complexity of the loss landscape through both the micro- and macro-level behaviors. This combination of theoretical richness and exact computability makes our bound (4) more practical and accurate.

For 0-1 loss and classifier $\boldsymbol{h}$, observe that $\Pr(\boldsymbol{h}(\boldsymbol{x}) \neq y) = F(P, \boldsymbol{h})$, which represents the (true) error of the prediction of $\boldsymbol{h}$ about an instance $(\boldsymbol{x}, y)$. Therefore, Theorem 3.2 implies the following.

**Corollary 1.** *Consider a classifier $\boldsymbol{h}$. Given the notations and i.i.d assumption as in Theorem 3.2, denote $\hat{a}_S = \Pr(\boldsymbol{h}(\boldsymbol{x}_s) \neq y_s | (\boldsymbol{x}_s, y_s) \in \boldsymbol{S})$ and $\hat{a}_i = \Pr(\boldsymbol{h}(\boldsymbol{x}_s) \neq y_s | (\boldsymbol{x}_s, y_s) \in \boldsymbol{S}_i)$ as the error of $\boldsymbol{h}$ on $\boldsymbol{S}_i$ for each index $i$. The following holds with probability at least $1 - \gamma^{-\alpha} - \delta$:*

$$\Pr(\boldsymbol{h}(\boldsymbol{x}) \neq y) \leq \hat{a}_S + \tfrac{b}{n} \sum_{i \in \boldsymbol{T}} \hat{a}_i + C\sqrt{\hat{u}\alpha \ln \gamma} + C\sqrt{\tfrac{\ln(2/\delta)}{2n}}$$

## 4 CONCENTRATION INEQUALITY FOR MULTINOMIAL RANDOM VARIABLES

The key innovative step to develop our error bounds before comes from the following novel result for multinomial random variables.

**Theorem 4.1.** *Consider $\boldsymbol{z} = (z_1, \ldots, z_K)$ which follows the multinomial distribution with parameters $n$ and $(p_1, ..., p_K) > 0$, and $a_1, ..., a_K$ are non-negative functions which may depend on $\boldsymbol{z}$. Let $\boldsymbol{T} = \{j : z_j > 0\}$, $a_o = 0$ if $|\boldsymbol{T}| = K$ and $(\sum_{k \notin \boldsymbol{T}} p_k)^{-1} \sum_{k \notin \boldsymbol{T}} a_k p_k$ otherwise. For any constant $\delta \in (0, 1]$, the following holds with probability at least $1 - \delta$:*

$$\sum_{i=1}^K a_i(p_i - \frac{z_i}{n}) \leq (a_o + \sum_{j \in \boldsymbol{T}} a_j)\sqrt{\frac{\ln(1/\delta)}{2n}} \tag{5}$$

This theorem suggests that a normalized multinomial variable $\frac{1}{n}(z_1, \ldots, z_K)$ should not be far from its expectation $(p_1, ..., p_K)$. With a high probability, the difference between the two is at most $gap = (a_o + \sum_{j \in \boldsymbol{T}} a_j)\sqrt{\ln(1/\delta)/(2n)}$, which represents the uncertainty. The proof for Theorem 4.1 requires us to derive novel bounds for binomial random variables. Some further results about both binomial and multinomial random variables can be found in Appendix B.

*Remark* 1. Several prior works have established bounds for multinomial random variables (Kawaguchi et al., 2022; Qian et al., 2020). In particular, Kawaguchi et al. (2022) replaced the right-hand side of (5) with $gap_K = (\hat{a}_o + \sqrt{2}\,\hat{a}_T)\sqrt{|\boldsymbol{T}| \ln(2K/\delta)/n} + \hat{a}_o 2|\boldsymbol{T}| \ln(2K/\delta)/n$, where $\hat{a}_o = \max_{j \notin \boldsymbol{T}} a_j$ and $\hat{a}_T = \max_{j \in \boldsymbol{T}} a_j$. The order of this gap is $O(\sqrt{\ln K/n})$, which represents a substantial improvement over the classical $O(\sqrt{K/n})$ bounds (Qian et al., 2020; Xu & Mannor, 2012). However, our bound improves further by an additional factor of $O(\sqrt{\ln K})$, making it tighter than the best prior results. This sharper rate is particularly valuable in high-dimensional settings.

*Remark* 2. It is also important to note that the bound $gap_K$ in (Kawaguchi et al., 2022) may perform poorly when the multinomial probabilities are comparable across categories (i.e., $p_i \approx p_j$). In such cases, $|\boldsymbol{T}|$ becomes large and approaches $K$ as $n$ grows, yielding $gap_K = O(\sqrt{K \ln K/n})$. This rate is in fact worse than the classical $O(\sqrt{K/n})$ bound. In contrast, our bound (5) may remain $O(1/\sqrt{n})$, when the term $(a_o + \sum_{j \in \boldsymbol{T}} a_j)$ is bounded by some constant or does not necessarily scale with $|\boldsymbol{T}|$ but instead depends more delicately on the actual magnitudes of the coefficients $a_i$. This highlights the robustness and superiority of our bound in challenging scenarios with nearly uniform multinomial distributions.

The remarks above highlight two key advantages of our bound over prior results. First, it achieves a tighter rate of $O(1/\sqrt{n})$, outperforming the best known $O(\sqrt{\ln K/n})$ rate (Kawaguchi et al., 2022), and significantly improving upon the classical $O(\sqrt{K/n})$ bounds (Qian et al., 2020; Xu & Mannor, 2012). Second, unlike earlier approaches, our bound remains stable even when the multinomial distribution is close to uniform, a regime where $gap_K$ deteriorates to $O(\sqrt{K \ln K/n})$ and thus becomes worse than the classical result. This robustness arises because our bound depends on the actual coefficients $a_i$, rather than scaling directly with the number of categories $K$ or the size of $\boldsymbol{T}$. Consequently, our result provides not only a sharper theoretical rate, but also a more reliable guarantee across a wide range of multinomial structures, including those with many categories or nearly uniform probabilities.

## 5 EVALUATION

In this section, we present an extensive evaluation for our bounds. We first verify tightness and nonvacuousness, and then other properties. More results appear in Appendix C.

### 5.1 ERROR GUARANTEES FOR LARGE-SCALE MODELS

Our first aim is to assess the theoretical guarantee for test error, without altering the pretrained models, using the training set only. We use 37 modern NNs[1] which were pretrained by Pytorch on the ImageNet dataset or COCO. Those models are 32 classifiers and 5 image segmentation models.

**Baselines:** The most closely related bounds are model-dependent, including those in (Kawaguchi et al., 2022; Than et al., 2025; von Luxburg & Bousquet, 2004; Hou et al., 2023; Bartlett et al., 2017; Arora et al., 2018; Golowich et al., 2020; Galanti et al., 2023b). However, these bounds are either *not computable* from the training set alone or are already known to be *vacuous* even for relatively small networks. Existing PAC-Bayes bounds (Biggs & Guedj, 2022; Lotfi et al., 2024a;b) typically require *significant modifications* to the target model, making them incompatible with our setting. Consequently, we adopt the bound of (Than & Phan, 2025) as the primary baseline for comparison.

**Setup:** We fix $\delta = 0.01, \alpha = 100, \gamma = 0.04^{-1/\alpha}$. This choice means that our bound is correct with a probability of at least 95%. We partition the data space by randomly initialized 200 centroids, and then assigning the training images into $K = 200$ regions using Euclidean distance. We use the 0-1 loss function to directly estimate the true classification error, as pointed out by Corollary 1. For this loss, we fix $C = 1$ as the worse case for simplicity.

**Results for large-scale classifiers:** Table 1 reports the results. It seems that a model with smaller training error can help our bound to obtain larger improvement margin. It is also worth observing that the estimates from our bound are at most 2 times greater than the test error of a model. In particular, our bound is tightest for "VIT L 16 V1". We attribute such a significant outperformance to our novel concentration results for multinomial random variables, which enable a small uncertainty term in bound (4).

**Results for semantic segmenters:** We use $K = 75$ due to the small size of the COCO dataset and the *mean IoU loss function* for evaluating the segmentation models. Table 2 shows the results. Across 5 models, our bound always performs better the prior one to provide tighter error estimates. One can be surprized by the large values of those error estimates. The main reason seems to come from the high training errors of those models. Another reason may be due to the number of training samples, which is much smaller than that of ImageNet. Nonetheless, we observe a strong correlation between the test errors and the estimates by our bound.

### 5.2 TRACKING THE DYNAMICS OF TEST ERROR OVER THE TRAINING COURSE

We next investigate *how well different parts in bound (4) can reflect the model performance*. To this end, we focus on the first two terms: $F(\boldsymbol{S}, \boldsymbol{h})$ and $mac_h = \sum_{i \in \boldsymbol{T}} F(\boldsymbol{S}_i, \boldsymbol{h})$ – a (macro) summary of the error from local areas. When $\boldsymbol{S}$ is a validation set, $F(\boldsymbol{S}, \boldsymbol{h})$ is the usual validation error.

**Setup:** We sample 20,000 images from the CIFAR10 dataset to train a ResNet18 architecture from scratch, while holding out 10,000 images as the validation set $\boldsymbol{S}$. For each epoch along the training process, we compute the accuracy on a separate test set, as well as $mac_h$ (for $K = 200$) and validation error from $\boldsymbol{S}$. The training is done three times to produce three different models with varying capacities.

**Result:** The results are reported in Figure 2. We observe that $mac_h$ exhibits a strong correlation to the true model performance, significantly better than both the training error and the validation error for the overfitting cases. Interestingly, the validation error continues increasing with more epochs while the test accuracy stabilizes. The correlation between validation error and test accuracy is positive, suggesting that validation error did not reflect well the true performance of the model. This could happen sometimes in practice. Furthermore, in the case of "Mild overfitting", while the

---

[1]https://pytorch.org/vision/stable/models.html

Table 1: Error estimates for large-scale **NN classifiers**. Each row show results for a model. The second column presents the model size, while the third and fourth column contain the training and test errors, respectively. The last two columns contain results from prior and our bounds on the true error (in %) of 32 NNs pretrained on ImageNet-1K.

| Model | #Params (M) | Training error | Test error | Error bound | |
|---|---|---|---|---|---|
| | | | | Than & Phan (2025) | Ours (4) |
| ResNet18 V1 | 11.7 | 21.24 | 30.31 | $57.90_{\pm 4.19}$ | $54.71_{\pm 2.64}$ |
| ResNet34 V1 | 21.8 | 15.67 | 26.75 | $52.32_{\pm 4.19}$ | $47.55_{\pm 2.65}$ |
| ResNet50 V1 | 25.6 | 13.12 | 23.88 | $49.77_{\pm 4.19}$ | $44.25_{\pm 2.60}$ |
| ResNet101 V1 | 44.5 | 10.50 | 22.68 | $47.15_{\pm 4.19}$ | $40.86_{\pm 2.64}$ |
| ResNet152 V1 | 60.2 | 10.13 | 21.78 | $46.78_{\pm 4.19}$ | $40.38_{\pm 2.65}$ |
| ResNet50 V2 | 25.6 | 8.94 | 19.18 | $45.59_{\pm 4.19}$ | $38.77_{\pm 2.62}$ |
| ResNet101 V2 | 44.5 | 6.01 | 18.19 | $42.66_{\pm 4.19}$ | $35.02_{\pm 2.61}$ |
| ResNet152 V2 | 60.2 | 5.18 | 17.77 | $41.83_{\pm 4.19}$ | $33.97_{\pm 2.60}$ |
| SwinTransformer B | 87.8 | 6.46 | 16.52 | $43.12_{\pm 4.19}$ | $35.65_{\pm 2.61}$ |
| SwinTransformer B V2 | 87.9 | 6.39 | 15.96 | $43.04_{\pm 4.19}$ | $35.54_{\pm 2.59}$ |
| SwinTransformer T | 28.3 | 9.99 | 18.59 | $46.64_{\pm 4.19}$ | $40.17_{\pm 2.62}$ |
| SwinTransformer T V2 | 28.4 | 8.72 | 18.02 | $45.38_{\pm 4.19}$ | $38.58_{\pm 2.64}$ |
| VGG13 | 133.0 | 18.46 | 30.11 | $55.11_{\pm 4.19}$ | $51.04_{\pm 2.69}$ |
| VGG13 BN | 133.1 | 19.22 | 28.53 | $55.87_{\pm 4.19}$ | $52.10_{\pm 2.66}$ |
| VGG19 | 143.7 | 16.12 | 27.70 | $52.77_{\pm 4.19}$ | $48.04_{\pm 2.64}$ |
| VGG19 BN | 143.7 | 15.94 | 25.76 | $52.59_{\pm 4.19}$ | $47.89_{\pm 2.69}$ |
| DenseNet121 | 8.0 | 15.63 | 25.63 | $52.28_{\pm 4.19}$ | $47.48_{\pm 2.63}$ |
| DenseNet161 | 28.7 | 10.48 | 22.96 | $47.13_{\pm 4.19}$ | $40.83_{\pm 2.64}$ |
| DenseNet169 | 14.1 | 12.40 | 24.36 | $49.05_{\pm 4.19}$ | $43.28_{\pm 2.62}$ |
| DenseNet201 | 20.0 | 9.81 | 23.10 | $46.46_{\pm 4.19}$ | $39.98_{\pm 2.66}$ |
| ConvNext Base | 88.6 | 5.21 | 16.07 | $41.86_{\pm 4.19}$ | $34.01_{\pm 2.60}$ |
| ConvNext Large | 197.8 | 3.85 | 15.63 | $40.50_{\pm 4.19}$ | $32.24_{\pm 2.58}$ |
| RegNet Y 128GF e2e | 644.8 | 5.57 | 11.78 | $42.22_{\pm 4.19}$ | $34.48_{\pm 2.61}$ |
| RegNet Y 128GF linear | 644.8 | 9.03 | 13.96 | $45.68_{\pm 4.19}$ | $39.00_{\pm 2.57}$ |
| RegNet Y 32GF e2e | 145.0 | 7.13 | 13.22 | $43.78_{\pm 4.19}$ | $36.48_{\pm 2.63}$ |
| RegNet Y 32GF linear | 145.0 | 10.56 | 15.48 | $47.21_{\pm 4.19}$ | $40.97_{\pm 2.63}$ |
| RegNet Y 32GF V2 | 145.0 | 3.76 | 16.78 | $40.41_{\pm 4.19}$ | $32.15_{\pm 2.61}$ |
| VIT B 16 linear | 86.6 | 14.97 | 18.05 | $51.62_{\pm 4.19}$ | $46.66_{\pm 2.62}$ |
| VIT B 16 V1 | 86.6 | 5.92 | 18.95 | $42.57_{\pm 4.19}$ | $34.91_{\pm 2.60}$ |
| VIT H 14 linear | 632.0 | 9.95 | 14.34 | $46.60_{\pm 4.19}$ | $40.16_{\pm 2.62}$ |
| VIT L 16 linear | 304.3 | 11.00 | 14.88 | $47.65_{\pm 4.19}$ | $41.55_{\pm 2.57}$ |
| VIT L 16 V1 | 304.3 | 3.47 | 20.40 | $40.12_{\pm 4.19}$ | $31.74_{\pm 2.59}$ |

Table 2: Error estimates for 5 **semantic segmentation models** pretrained on COCO 2017. Each row show results for a model. The last two columns show prior and our bounds on the true error (in %).

| Model | #Params (M) | Train error | Test error | Error bound | |
|---|---|---|---|---|---|
| | | | | Than & Phan (2025) | Ours (4) |
| DeepLabV3_MobileNet_V3 | 11.0 | 35.51 | 38.01 | $86.25_{\pm 0.53}$ | $81.17_{\pm 0.56}$ |
| FCN_ResNet50 | 35.3 | 38.20 | 41.32 | $88.94_{\pm 0.53}$ | $85.10_{\pm 0.60}$ |
| FCN_ResNet101 | 54.3 | 32.39 | 35.76 | $83.13_{\pm 0.53}$ | $77.05_{\pm 0.65}$ |
| DeepLabV3_ResNet50 | 42.0 | 28.65 | 31.34 | $79.39_{\pm 0.53}$ | $71.90_{\pm 0.57}$ |
| DeepLabV3_ResNet101 | 61.0 | 27.16 | 30.60 | $77.90_{\pm 0.53}$ | $69.85_{\pm 0.57}$ |

model performs quite well, validation error did not detect this good performance. Those behaviors suggest that *sometimes validation error may not be a good indicator for model performance.*

Surprizingly, $mac_h$ can reflect the model performance very well. For non-overfitting cases, $mac_h$ is quite stable when the training error becomes stable and exhibits a near-perfect correlation to test accuracy. For the overfitting cases, $mac_h$ behaves unstable even though the training error stabilizes around zero. Furthermore, $mac_h$ is still high in the ending epochs. Those behaviors also appear for different datasets and architectures, as reported in Appendix C.3. They suggest that $mac_h$ *can accurately track the dynamic of the model performance*, which is practically beneficial.

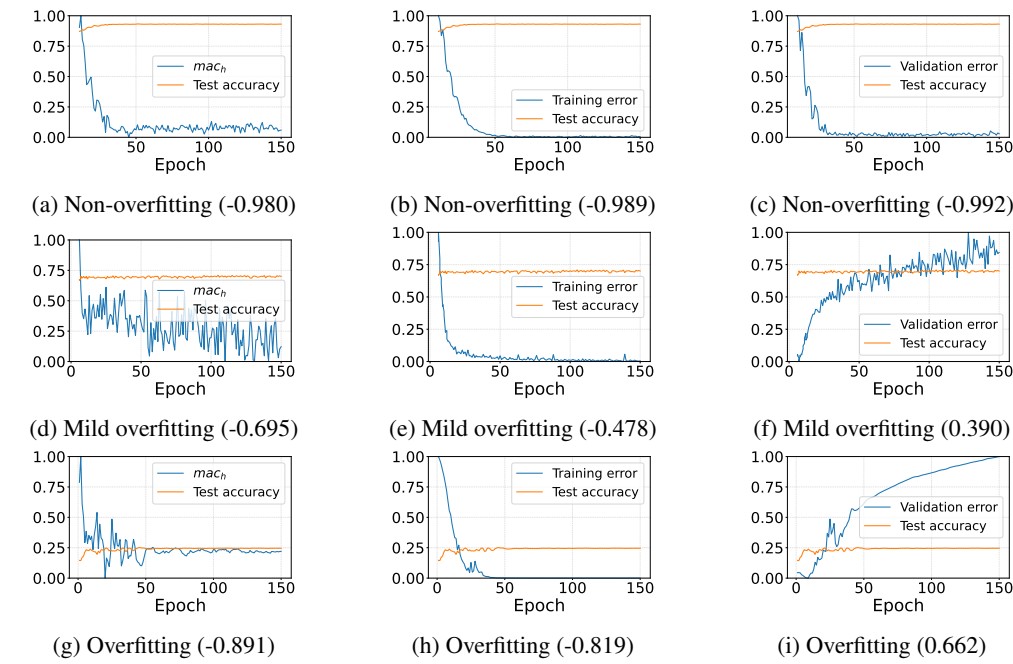

(a) Non-overfitting (-0.980)     (b) Non-overfitting (-0.989)     (c) Non-overfitting (-0.992)

(d) Mild overfitting (-0.695)     (e) Mild overfitting (-0.478)     (f) Mild overfitting (0.390)

(g) Overfitting (-0.891)     (h) Overfitting (-0.819)     (i) Overfitting (0.662)

Figure 2: Correlation of $mac_h$, training error, and validation error (respectively) to test accuracy during training **ResNet18** on **CIFAR10**. Each row corresponds to a different trained model. The correlation coefficient is shown in brackets - a lower correlation value is better.

### 5.3 TIGHT BOUND FOR MULTINOMIAL VARIABLES

Finally, we want to investigate the tightness of our bound (5) for a multinomial random variable in a realistic setting. Therefore we compare our bound, denoted as $gap_O = (1 + \sum_{j \in \boldsymbol{T}} f_j)\sqrt{\frac{\ln(1/\delta)}{2n}}$ and the one in (Kawaguchi et al., 2022), denoted as $gap_K = (1 + \sqrt{2}\,\hat{a}_T)\sqrt{\frac{|\boldsymbol{T}|\ln(2K/\delta)}{n}} + \frac{2|\boldsymbol{T}|\ln(2K/\delta)}{n}$, where $f_i = F(\boldsymbol{S}_i, \boldsymbol{h})$ comes from a classifier $\boldsymbol{h}$ and 0-1 loss, $\hat{a}_T = \max_{i \in \boldsymbol{T}} f_i$. Note that $a_o$ in bound (5) is upper bounded by 1 and the intractable term $a_j$ is approximated by $f_j$.

The ImageNet training set and 32 pretrained PyTorch classification models are used for this evaluation, with $K = 200$. For each model, we compute $gap_O$ and $gap_K$ five times. The results are reported in Figure 8 in Appendix C.6. The results tell that $gap_O$ outperforms $gap_K$ in all 32 cases, sometimes with a significant margin. We observe that an accurate model $\boldsymbol{h}$ (e.g., ConvNext Large and VIT L 16 V1) often leads to a much smaller $gap_O$, due to many small local error $f_j$. A model with a high error can cause both $gap_O$ and $gap_K$ to be high. This empirical behavior well supports the theoretical discussion in Section 4. The superiority of $gap_O$ over $gap_K$ also appears when changing $K$, as reported by Figure 9 in Appendix C.6 for some pretrained ImageNet classifiers.

## 6 CONCLUSION

In this work, we introduce novel bounds on the test error of trained models. These bounds possess several desirable properties and address key limitations of prior approaches. In particular, they enable the tracking of a model's generalization dynamics, offering practical utility for model evaluation and diagnosis.

Nevertheless, several limitations remain. First, our bounds can become vacuous in the small-sample regime, where the third term in bound (4) may dominate. Second, while the constant $C$ can be bounded for common loss functions (e.g., 0–1, absolute, hinge, and squared loss), certain losses yield infinite values, which may cause the bound vacuous. Finally, our results are model-dependent, which may limit their applicability in settings where the goal is to analyze learning algorithms or hypothesis classes. These limitations highlight important directions for future research.

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

## A  PROOFS

### A.1  FOR MAIN THEOREMS

*Proof of Theorem 3.1.*  Firstly, we observe that

$$F(P, \boldsymbol{h}) - \sum_{i=1}^{K} \frac{n_i}{n} a_i(\boldsymbol{h}) = \sum_{i=1}^{K} P(\mathcal{Z}_i) a_i(\boldsymbol{h}) - \sum_{i=1}^{K} \frac{n_i}{n} a_i(\boldsymbol{h}) = \sum_{i=1}^{K} a_i(\boldsymbol{h}) \left[ p_i - \frac{n_i}{n} \right]$$

Note that $(n_1, ..., n_K)$ is a multinomial random variable with parameters $n$ and $(p_1, ..., p_K)$. Each $n_i$ is a binomial random variable with parameters $n$ and $p_i$. Furthermore $a_i(\boldsymbol{h}) \geq 0$ for all $i$. Therefore, according to Theorem B.5, we have:

$$\Pr\left( \sum_{i=1}^{K} a_i(\boldsymbol{h}) \left[ p_i - \frac{n_i}{n} \right] \geq \sum_{i \in \boldsymbol{T}} a_i(\boldsymbol{h}) \sqrt{\frac{-\ln \delta_1}{2n}} + a_o \sqrt{\frac{-\ln \delta_1}{2n}} \right) \leq \delta_1$$

Therefore

$$\Pr\left( F(P, \boldsymbol{h}) - \sum_{i=1}^{K} \frac{n_i}{n} a_i(\boldsymbol{h}) \geq \sum_{i \in \boldsymbol{T}} a_i(\boldsymbol{h}) \sqrt{\frac{-\ln \delta_1}{2n}} + a_o \sqrt{\frac{-\ln \delta_1}{2n}} \right) \leq \delta_1 \qquad (6)$$

A rearrangment leads to

$$\Pr\left( F(P, \boldsymbol{h}) \geq \frac{1}{n} \sum_{i \in \boldsymbol{T}} (n_i + b) a_i(\boldsymbol{h}) + a_o \sqrt{\frac{-\ln \delta_1}{2n}} \right) \leq \delta_1 \qquad (7)$$

On the other hand, Theorem A.1 below shows that

$$\Pr\left( \frac{1}{n} \sum_{i \in \boldsymbol{T}} (n_i + b) a_i(\boldsymbol{h}) \geq \frac{1}{n} \sum_{i \in \boldsymbol{T}} (n_i + b) F(\boldsymbol{S}_i, \boldsymbol{h}) + C \sqrt{\frac{u}{2n^2} \ln \frac{1}{\delta_2}} \right) \leq \delta_2 \qquad (8)$$

Combining this with (7) and the union bound, we have

$$\Pr\left( F(P, \boldsymbol{h}) \geq \frac{1}{n} \sum_{i \in \boldsymbol{T}} (n_i + b) F(\boldsymbol{S}_i, \boldsymbol{h}) + C \sqrt{\frac{u}{2n^2} \ln \frac{1}{\delta_2}} + a_o \sqrt{\frac{-\ln \delta_1}{2n}} \right) \leq \delta_1 + \delta_2 \qquad (9)$$

In other words

$$\Pr\left( F(P, \boldsymbol{h}) \geq F(\boldsymbol{S}, \boldsymbol{h}) + \frac{b}{n} \sum_{i \in \boldsymbol{T}} F(\boldsymbol{S}_i, \boldsymbol{h}) + C \sqrt{\frac{u}{2n^2} \ln \frac{1}{\delta_2}} + a_o \sqrt{\frac{-\ln \delta_1}{2n}} \right) \leq \delta_1 + \delta_2 \qquad (10)$$

completing the proof.  □

*Proof of Theorem 3.2.*  Theorem 3.1 shows that, for any $\delta_2 \geq \exp(-\frac{u \ln \gamma}{\max\{4b, 8n-6\}})$,

$$\Pr\left( F(P, \boldsymbol{h}) \geq F(\boldsymbol{S}, \boldsymbol{h}) + \frac{b}{n} \sum_{i=1}^{K} F(\boldsymbol{S}_i, \boldsymbol{h}) + C \sqrt{\frac{u}{2n^2} \ln \frac{1}{\delta_2}} + a_o \sqrt{\frac{\ln(2/\delta)}{2n}} \right) \leq \delta/2 + \delta_2 \qquad (11)$$

By the definition of $p_i$, one can easily show that $\sum_{i=1}^{K} p_i^2$ is minimized at $1/K$. As a result, $u = \gamma n(1+2b) + |\boldsymbol{T}| b^2 + \gamma^2 n^2 (\sum_{i=1}^{K} p_i^2) \geq \gamma n(1+2b) + |\boldsymbol{T}| b^2 + \gamma^2 n^2 / K$ and $\exp(-\frac{u \ln \gamma}{\max\{4b, 8n-6\}}) \leq \exp(-\frac{\gamma n(1+2b) + |\boldsymbol{T}| b^2 + \gamma^2 n^2 / K}{\max\{4b, 8n-6\}} \ln \gamma) \leq \gamma^{-\alpha}$. By choosing $\delta_2 = \gamma^{-\alpha}$, we can rewrite (11) as

$$\Pr\left( F(P, \boldsymbol{h}) \geq F(\boldsymbol{S}, \boldsymbol{h}) + \frac{b}{n} \sum_{i=1}^{K} F(\boldsymbol{S}_i, \boldsymbol{h}) + C \sqrt{\frac{u}{2n^2} \alpha \ln \gamma} + a_o \sqrt{\frac{\ln(2/\delta)}{2n}} \right) \leq \delta/2 + \gamma^{-\alpha} \qquad (12)$$

Since $(n_1, ..., n_K)$ is a multinomial random variable with parameters $n$ and $(p_1, ..., p_K)$, Lemma B.6 shows that $\Pr\left(\sum_{i=1}^{K} p_i^2 \geq \sum_{i=1}^{K} \left(\frac{n_i}{n}\right)^2 + \sqrt{\frac{2\ln(2/\delta)}{n}}\right) \leq \delta/2$. Therefore

$$\Pr\left(\frac{u}{2n^2} \geq \frac{\gamma(1+2b)}{2n} + \frac{\gamma|\boldsymbol{T}|b^2}{2n^2} + \frac{\gamma^2}{2}\sum_{i=1}^{K}\left(\frac{n_i}{n}\right)^2 + \gamma^2\sqrt{\frac{\ln(2/\delta)}{2n}}\right) \leq \delta/2 \tag{13}$$

Combining (12), (13) and the union bound leads to the following

$$\Pr\left(F(P, \boldsymbol{h}) \geq F(\boldsymbol{S}, \boldsymbol{h}) + \frac{b}{n}\sum_{i=1}^{K} F(\boldsymbol{S}_i, \boldsymbol{h}) + C\sqrt{\hat{u}\alpha\ln\gamma} + a_o\sqrt{\frac{\ln(2/\delta)}{2n}}\right) \leq \gamma^{-\alpha} + \delta \tag{14}$$

By the definition of $a_o$, observe that $a_o \leq (\sum_{k \notin \boldsymbol{T}} p_k)^{-1} \sum_{k \notin \boldsymbol{T}} p_k a_k(\boldsymbol{h})$, which is a convex combination of all $a_k(\boldsymbol{h})$ for $k \notin \boldsymbol{T}$. Therefore $a_o \leq C$, due to $a_k(\boldsymbol{h}) \leq C, \forall k$. As a result

$$\Pr\left(F(P, \boldsymbol{h}) \geq F(\boldsymbol{S}, \boldsymbol{h}) + \frac{b}{n}\sum_{i=1}^{K} F(\boldsymbol{S}_i, \boldsymbol{h}) + C\sqrt{\hat{u}\alpha\ln\gamma} + C\sqrt{\frac{\ln(2/\delta)}{2n}}\right) \leq \delta + \gamma^{-\alpha} \tag{15}$$

completing the proof. $\qquad\square$

### A.2 Approximating local errors by a data set

**Theorem A.1.** *Given the notations in Theorem 3.1,*

$$\Pr\left(\sum_{i \in \boldsymbol{T}} \frac{n_i + b}{n} a_i(\boldsymbol{h}) \geq \sum_{i \in \boldsymbol{T}} \frac{n_i + b}{n} F(\boldsymbol{S}_i, \boldsymbol{h}) + C\sqrt{\frac{u}{2n^2}\ln\frac{1}{\delta_2}}\right) \leq \delta_2 \tag{16}$$

$$\Pr\left(\sum_{i \in \boldsymbol{T}} \frac{n_i + b}{n} F(\boldsymbol{S}_i, \boldsymbol{h}) \geq \sum_{i \in \boldsymbol{T}} \frac{n_i + b}{n} a_i(\boldsymbol{h}) + C\sqrt{\frac{u}{2n^2}\ln\frac{1}{\delta_2}}\right) \leq \delta_2 \tag{17}$$

*Proof.* Denote $\boldsymbol{n} = \{n_1, ..., n_K\}$ and for each $j \in [K]$:

$$B_j = \sum_{i=1}^{j}(n_i + b_i)a_i(\boldsymbol{h}) - \sum_{i=1}^{j}(n_i + b_i)F(\boldsymbol{S}_i, \boldsymbol{h}) \tag{18}$$

$$X_j = (n_j + b_j)F(\boldsymbol{S}_j, \boldsymbol{h}) \tag{19}$$

$$b_j = \begin{cases} b & \text{if } j \in \boldsymbol{T}, \\ 0 & \text{otherwise} \end{cases} \tag{20}$$

$$\boldsymbol{S}_{\leq j} = \bigcup_{i \leq j} \boldsymbol{S}_i \tag{21}$$

Denote $y = \frac{4t}{uC^2}$ for any $t \in \left[0, uC\sqrt{\phi/2}\right]$, where $\phi = \min\{\frac{\ln\gamma}{\max\{4b_1, 8n-6\}}, ..., \frac{\ln\gamma}{\max\{4b_K, 8n-6\}}\}$. The proof for (16) contains three main steps.

**Step 1:** We first observe that

$$\Pr\left(B_K \geq t\right) \leq e^{-yt}\mathbb{E}_{\boldsymbol{S}}\left[e^{yB_K}\right] \qquad \text{(Chernoff bounds)} \tag{22}$$

$$\leq e^{-yt}\mathbb{E}_{\boldsymbol{h}, \boldsymbol{n}}\left[\mathbb{E}_{\boldsymbol{S}}\left[e^{yB_K}|\boldsymbol{h}, \boldsymbol{n}\right]\right] \qquad \text{(Law of total expectation)} \tag{23}$$

**Step 2 - estimating** $\mathbb{E}_{\boldsymbol{S}}\left[e^{yB_K}|\boldsymbol{h}, \boldsymbol{n}\right]$**:** We observe the following for each $j \in \boldsymbol{T}$,

$$\mathbb{E}_{X_j}[X_j|\boldsymbol{h}, \boldsymbol{n}] = \mathbb{E}_{\boldsymbol{S}_j}[(n_j + b_j)F(\boldsymbol{S}_j, \boldsymbol{h})|\boldsymbol{h}, \boldsymbol{n}] \tag{24}$$

$$= (n_j + b_j)\mathbb{E}_{\boldsymbol{S}_j}\left[\frac{1}{n_j}\sum_{i=1}^{n_j}\ell(\boldsymbol{h}, \boldsymbol{z}_{ji})|\boldsymbol{h}, \boldsymbol{n}\right] \qquad \text{(where } \boldsymbol{S}_j = \{\boldsymbol{z}_{ji}\}_{i=1}^{n_j}) \tag{25}$$

$$= \frac{n_j + b_j}{n_j}\sum_{i=1}^{n_j}\mathbb{E}_{\boldsymbol{z}_{ji} \in \mathcal{Z}_j}[\ell(\boldsymbol{h}, \boldsymbol{z}_{ji})|\boldsymbol{h}, \boldsymbol{n}] \qquad (\boldsymbol{S}_j \text{ contains i.i.d. samples in } \mathcal{Z}_j) \tag{26}$$

$$= \frac{n_j + b_j}{n_j}\sum_{i=1}^{n_j}a_j(\boldsymbol{h}) = (n_j + b_j)a_j(\boldsymbol{h}) \tag{27}$$

Therefore $B_j = B_{j-1} + \mathbb{E}_{X_j}[X_j|\boldsymbol{h}, \boldsymbol{n}] - X_j$ for all $j \in \boldsymbol{T}$. Note that $B_i = B_{i-1}$ (due to $n_i = b_i = X_i = 0$) for all $i \notin \boldsymbol{T}$. Hence, for $i \notin \boldsymbol{T}$, we will use $\mathbb{E}_{X_i}[X_i|\boldsymbol{h}, \boldsymbol{n}] - X_i$ instead of 0 in the below analysis for simplicity of presentation.

We can rewrite

$$\mathbb{E}_{\boldsymbol{S}}\left[e^{yB_K}|\boldsymbol{h}, \boldsymbol{n}\right] = \mathbb{E}_{\boldsymbol{S}}\left[e^{y(B_{K-1}+\mathbb{E}_{X_K}[X_K|\boldsymbol{h}, \boldsymbol{n}]-X_K)}|\boldsymbol{h}, \boldsymbol{n}\right] \tag{28}$$

$$= \mathbb{E}_{\boldsymbol{S}_{\leq K}}\left[e^{y(B_{K-1}+\mathbb{E}_{X_K}[X_K|\boldsymbol{h}, \boldsymbol{n}]-X_K)}|\boldsymbol{h}, \boldsymbol{n}\right] \tag{29}$$

$$\leq \mathbb{E}_{\boldsymbol{S}_{\leq K-1}}\left[e^{yB_{K-1}}|\boldsymbol{h}, \boldsymbol{n}\right]\mathbb{E}_{X_K}\left[e^{y(\mathbb{E}_{X_K}[X_K|\boldsymbol{h}, \boldsymbol{n}]-X_K)}|\boldsymbol{h}, \boldsymbol{n}\right] \tag{30}$$

where the last inequality comes from the fact that $X_K$ is conditionally independent with $\boldsymbol{S}_{\leq K-1}$, conditioned on $\{\boldsymbol{h}, \boldsymbol{n}\}$.

It is easy to see that $0 \leq X_K \leq C(n_K + b_K)$, due to $0 \leq F(\boldsymbol{S}_K, \boldsymbol{h}) \leq C$. Lemma B.1 implies $\mathbb{E}_{X_K}\left[e^{y(\mathbb{E}_{X_K}[X_K|\boldsymbol{h}, \boldsymbol{n}]-X_K)}|\boldsymbol{h}, \boldsymbol{n}\right] \leq \exp\left(\frac{y^2 C^2 (n_K + b_K)^2}{8}\right)$. Plugging this into (30), we obtain

$$\mathbb{E}_{\boldsymbol{S}}\left[e^{yB_K}|\boldsymbol{h}, \boldsymbol{n}\right] \leq \mathbb{E}_{\boldsymbol{S}_{\leq K-1}}\left[e^{yB_{K-1}}|\boldsymbol{h}, \boldsymbol{n}\right]\exp\left(\frac{y^2 C^2 (n_K + b_K)^2}{8}\right) \tag{31}$$

Using the same arguments for $X_{K-1}, ..., X_1$, we obtain the followings

$$\mathbb{E}_{\boldsymbol{S}}\left[e^{yB_K}|\boldsymbol{h}, \boldsymbol{n}\right] \leq \mathbb{E}_{\boldsymbol{S}_{\leq K-2}}\left[e^{yB_{K-2}}|\boldsymbol{h}, \boldsymbol{n}\right]\exp\left(\frac{y^2 C^2 (n_K + b_K)^2}{8} + \frac{y^2 C^2 (n_{K-1} + b_{K-1})^2}{8}\right)$$

$$...$$

$$\leq \exp\left(\frac{y^2 C^2}{8}\sum_{i=1}^{K}(n_i + b_i)^2\right) \tag{32}$$

**Step 3 - bounding** $\Pr(B_K \geq t)$**:** By combining (32) with (23), we obtain

$$\Pr(B_K \geq t) \leq e^{-yt}\mathbb{E}_{\boldsymbol{h}, \boldsymbol{n}}\exp\left(\frac{y^2 C^2}{8}\sum_{i=1}^{K}(n_i + b_i)^2\right) \tag{33}$$

$$= e^{-yt}\mathbb{E}_{\boldsymbol{n}}\exp\left(\frac{y^2 C^2}{8}\sum_{i=1}^{K}(n_i + b_i)^2\right) \tag{34}$$

$$\leq e^{-yt}\mathbb{E}_{\boldsymbol{n}}\exp\left(\frac{y^2 C^2}{8}\sum_{i=1}^{K-1}(n_i + b_i)^2\right)\mathbb{E}_{n_K}\exp\left(\frac{y^2 C^2}{8}(n_K + b_K)^2\right) \tag{35}$$

(Since $n_K$ is independent with $n_1, ..., n_{K-1}$)

Since $t \in \left[0, uC\sqrt{\phi/2}\right]$, observe that $\frac{y^2 C^2}{8} = \frac{2t^2}{u^2 C^2} \leq \phi \leq \frac{\ln\gamma}{\max\{4b_K, 8n-6\}} \leq \min\{\frac{\ln\gamma}{4b_K}, \frac{\ln\gamma}{2(4n-3)}\} \leq \min\{\frac{\ln\gamma}{4b_K(1-\gamma p_K)}, \frac{\ln\gamma}{2(1-\gamma p_K)(4n-3)}\}$ when $\gamma p_K < 1$. Note that $n_K$ is a binomial random variable with parameters $n$ and $p_K$. Combining those facts with Lemma B.4

implies that $\mathbb{E}_{n_K} \exp\left(\frac{y^2 C^2}{8}(n_K + b_K)^2\right) \leq \exp\left(\frac{y^2 C^2}{8}\left((1 + \gamma n p_K)\gamma n p_K + 2\gamma n p_K b_K + b_K^2\right)\right)$.
The same result also holds when $\gamma p_K \geq 1$. As a result, those facts and (35) lead to the following:

$$\Pr\left(B_K \geq t\right) \leq e^{-yt}\mathbb{E}_{\boldsymbol{n}} \exp\left(\frac{y^2 C^2}{8}\sum_{i=1}^{K-1}(n_i + b_i)^2\right)\exp\left(\frac{y^2 C^2}{8}\left((1 + \gamma n p_K)\gamma n p_K + 2\gamma n p_K b_K + b_K^2\right)\right) \tag{36}$$

Using the same arguments for the remaining variables $n_{K-1}, ..., n_1$, we obtain

$$\begin{aligned}
\Pr\left(B_K \geq t\right) &\leq \exp\left(-yt + \frac{y^2 C^2}{8}\sum_{i=1}^{K}\left((1 + \gamma n p_i)\gamma n p_i + 2\gamma n p_i b_i + b_i^2\right)\right) \tag{37}\\
&\leq \exp\left(-yt + \frac{y^2 C^2}{8}\left(\sum_{i=1}^{K}\gamma^2 n^2 p_i^2 + \gamma n + 2\gamma n b\sum_{i \in \boldsymbol{T}}p_i + |\boldsymbol{T}|b^2\right)\right) \tag{38}\\
&\qquad \left(\text{Due to } \sum_{i=1}^{K}p_i = 1\right) \\
&\leq \exp\left(-yt + \frac{y^2 C^2}{8}\left(\sum_{i=1}^{K}\gamma^2 n^2 p_i^2 + \gamma n(1 + 2b) + |\boldsymbol{T}|b^2\right)\right) \tag{39}\\
&= \exp\left(-yt + \frac{y^2 C^2 u}{8}\right) = \exp\left(\frac{-2t^2}{uC^2}\right) \tag{40}
\end{aligned}$$

As a result

$$\Pr\left(\sum_{i=1}^{K}(n_i + b_i)a_i(\boldsymbol{h}) \geq \sum_{i=1}^{K}(n_i + b_i)F(\boldsymbol{S}_i, \boldsymbol{h}) + t\right) \leq \exp\left(-\frac{2t^2}{uC^2}\right) \tag{41}$$

Since $n_j = 0$ and $b_j = 0$ for all $j \notin \boldsymbol{T}$, we have

$$\Pr\left(\sum_{i \in \boldsymbol{T}}(n_i + b)a_i(\boldsymbol{h}) \geq \sum_{i \in \boldsymbol{T}}(n_i + b)F(\boldsymbol{S}_i, \boldsymbol{h}) + t\right) \leq \exp\left(-\frac{2t^2}{uC^2}\right) \tag{42}$$

Multiplying both sides (of the probability term) with $1/n$ leads to

$$\Pr\left(\sum_{i \in \boldsymbol{T}}\frac{n_i + b}{n}a_i(\boldsymbol{h}) \geq \sum_{i \in \boldsymbol{T}}\frac{n_i + b}{n}F(\boldsymbol{S}_i, \boldsymbol{h}) + t/n\right) \leq \exp\left(-\frac{2t^2}{uC^2}\right)$$

Choosing $t = C\sqrt{\frac{u}{2}\ln\frac{1}{\delta_2}}$ results in (16).

The second result can be proven by considering $A_K = \sum_{i=1}^{K}(n_i + b_i)F(\boldsymbol{S}_i, \boldsymbol{h}) - \sum_{i=1}^{K}(n_i + b_i)a_i(\boldsymbol{h})$ and bounding $\Pr\left(A_K \geq t\right)$ using the same arguments above, completing the proof. $\square$

# B SUPPORTING THEOREMS AND LEMMAS

## B.1 HOEFFDING'S LEMMA FOR CONDITIONALS

**Lemma B.1** (Hoeffding's lemma for conditionals). *Let $X$ be any real-valued random variable that may depend on some random variables $\boldsymbol{Y}$. Assume that $a \leq X \leq b$ almost surely, for some constants $a, b$. Then, for all $\lambda \in \mathbb{R}$,*

$$\mathbb{E}_X\left[e^{\lambda(\mathbb{E}_X[X|\boldsymbol{Y}]-X)}|\boldsymbol{Y}\right] \leq \exp\left(\frac{\lambda^2(b-a)^2}{8}\right) \tag{43}$$

$$\mathbb{E}_X\left[e^{\lambda(X-\mathbb{E}_X[X|\boldsymbol{Y}])}|\boldsymbol{Y}\right] \leq \exp\left(\frac{\lambda^2(b-a)^2}{8}\right) \tag{44}$$

*Proof.* Denote $c = \mathbb{E}_X[X|\boldsymbol{Y}] - b, d = \mathbb{E}_X[X|\boldsymbol{Y}] - a$ and hence $c \leq 0 \leq d$.

Since $\exp$ is a convex function, we have the following for all $\mathbb{E}_X[X|\boldsymbol{Y}] - X \in [c, d]$:

$$e^{\lambda(\mathbb{E}_X[X|\boldsymbol{Y}]-X)} \leq \frac{d - \mathbb{E}_X[X|\boldsymbol{Y}] + X}{d - c} e^{\lambda c} + \frac{\mathbb{E}_X[X|\boldsymbol{Y}] - X - c}{d - c} e^{\lambda d}$$

Therefore, by taking the conditional expectation over $X$ for both sides,

$$\mathbb{E}_X\left[e^{\lambda(\mathbb{E}_X[X|\boldsymbol{Y}]-X)}\Big|\boldsymbol{Y}\right] \leq \frac{d - \mathbb{E}_X[X|\boldsymbol{Y}] + \mathbb{E}_X[X|\boldsymbol{Y}]}{d - c} e^{\lambda c} + \frac{\mathbb{E}_X[X|\boldsymbol{Y}] - \mathbb{E}_X[X|\boldsymbol{Y}] - c}{d - c} e^{\lambda d}$$

$$= \frac{d}{d - c} e^{\lambda c} - \frac{c}{d - c} e^{\lambda d} \tag{45}$$

$$= e^{L(\lambda(d-c))} \tag{46}$$

where $L(h) = \frac{ch}{d-c} + \ln(1 + \frac{c - e^h c}{d-c})$. For this function, note that

$$L(0) = L'(0) = 0 \text{ and } L''(h) = -\frac{cde^h}{(d - ce^h)^2}$$

The AM-GM inequality suggests that $L''(h) \leq 1/4$ for all $h$. Combining this property with Taylor's theorem leads to the following, for some $\theta \in [0, 1]$,

$$L(h) = L(0) + hL'(0) + \frac{1}{2}h^2 L''(h\theta) \leq \frac{h^2}{8}$$

Combining this with (46) leads to the first statement.

The second result can be proven by using the same arguments as above for $X - \mathbb{E}_X[X|\boldsymbol{Y}]$, completing the proof. $\qquad\square$

## B.2 BINOMIAL RANDOM VARIABLES

We next present some properties of binomial random variables.

**Lemma B.2.** *Consider a binomial random variable $z$ with parameters $n \geq 1$ and $\eta \in [0, 1]$. For all $\epsilon \geq 0$:*

$$\Pr(n\eta - z \geq n\epsilon) \leq e^{-2n\epsilon^2} \tag{47}$$

$$\Pr(z - n\eta \geq n\epsilon) \leq e^{-2n\epsilon^2} \tag{48}$$

*Proof.* By definition, we can rewrite $z = \sum_{i=1}^n x_i$, where $x_1, ..., x_n$ are i.i.d. samples from the Bernoulli distribution with parameter $\eta$. We can write $n\eta - z = \sum_{i=1}^n (\mathbb{E}[x_i] - x_i)$ and therefore

$$\mathbb{E}e^{4\epsilon(n\eta-z)} = \mathbb{E}\exp\left(4\epsilon \sum_{i=1}^n (\mathbb{E}[x_i] - x_i)\right) \tag{49}$$

Note that $\mathbb{E}[e^{X+Y}] = \mathbb{E}[e^X]\mathbb{E}[e^Y]$ when $X$ and $Y$ are independent. Combining this fact with the independence of $\{x_1, \ldots, x_n\}$ leads to the following:

$$\mathbb{E}e^{4\epsilon(n\eta-z)} = \mathbb{E}\exp\left(4\epsilon \sum_{i=1}^n (\mathbb{E}[x_i] - x_i)\right) = \prod_{i=1}^n \mathbb{E}[e^{4\epsilon(\mathbb{E}[x_i]-x_i)}] \tag{50}$$

Observe further that $0 \leq x_i \leq 1$. Hoeffding's lemma implies $\mathbb{E}e^{4\epsilon(\mathbb{E}[x_i]-x_i)} \leq e^{\frac{(4\epsilon)^2}{8}} = e^{2\epsilon^2}$ for all $i$. Combining this fact with (50), we obtain

$$\mathbb{E}e^{4\epsilon(n\eta-z)} \leq e^{2n\epsilon^2} \tag{51}$$

Using this result with Chernoff bounds, we obtain the following:

$$\Pr(n\eta - z \geq n\epsilon) \leq e^{-4n\epsilon^2} \mathbb{E}e^{4\epsilon(n\eta-z)} \leq e^{-4n\epsilon^2} e^{2n\epsilon^2} = e^{-2n\epsilon^2} \tag{52}$$

We can use the same arguments to show the other results, completing the proof. $\qquad\square$

**Lemma B.3.** *Consider a binomial random variable $z$ with parameters $n \geq 1$ and $\eta \in [0, 1]$. For any $c \geq 1$,*

- $\mathbb{E}e^{\lambda z} \leq e^{cn\eta\lambda}$, *for all $\lambda \geq 0$, if $c\eta \geq 1$.*
- $\mathbb{E}e^{\lambda z} \leq e^{cn\eta\lambda}$, *for all $\lambda \in [0, \frac{\ln c}{1-c\eta}]$, if $c\eta < 1$.*

*Proof.* Since $z$ is a binomial random variable, we can write $z = x_1 + \cdots + x_n$, where $x_1, ..., x_n$ are i.i.d. Bernoulli random variables with parameter $\eta$.

Consider the case $c\eta \geq 1$. Since $x_1, ..., x_n$ are i.i.d., we have $\mathbb{E}e^{\lambda(x_i + x_j)} = \mathbb{E}e^{\lambda x_i}\mathbb{E}e^{\lambda x_j}$ for any $i, j$. Therefore applying Lemma 5 in (Than & Phan, 2025), we have

$$\mathbb{E}e^{\lambda z} = \mathbb{E}e^{\lambda(x_1 + \cdots + x_n)} = \mathbb{E}e^{\lambda x_1} \cdots \mathbb{E}e^{\lambda x_n} \leq e^{c\eta\lambda} \cdots e^{c\eta\lambda} = e^{nc\eta\lambda}$$

We can show the same result for the case $c\eta < 1$, which completes the proof. $\square$

**Lemma B.4.** *Consider a binomial random variable $z$ with parameters $n \geq 1$ and $\eta \in [0, 1]$. For all constants $b \geq 0, c \geq 1$:*

- $\mathbb{E}e^{\lambda(z+b)^2} \leq e^{cn\eta(1+cn\eta)\lambda + 2cn\eta b\lambda + b^2\lambda}$, *for all $\lambda \geq 0$, if $c\eta \geq 1$.*
- $\mathbb{E}e^{\lambda(z+b)^2} \leq e^{cn\eta(1+cn\eta)\lambda + 2cn\eta b\lambda + b^2\lambda}$, *for all $\lambda \in [0, d]$ with $d = \min\{\frac{\ln c}{4b(1-c\eta)}, \frac{\ln c}{2(1-c\eta)(4n-3)}\}$, if $c\eta < 1$.*

*Proof.* Observe that

$$\mathbb{E}e^{\lambda(z+b)^2} = \mathbb{E}e^{\lambda(z^2+2bz+b^2)} \leq \sqrt{\mathbb{E}e^{2\lambda z^2}}\sqrt{\mathbb{E}e^{2\lambda(2bz+b^2)}} \tag{53}$$

where we have used Holder's inequality. When $c\eta < 1$ and $\lambda \in [0, d]$, Lemma 7 in (Than & Phan, 2025) shows that $\mathbb{E}e^{2\lambda z^2} \leq e^{2cn\eta(1+cn\eta)\lambda}$ and Lemma B.3 implies $\mathbb{E}e^{2\lambda(2bz+b^2)} \leq e^{2\lambda(2cn\eta b+b^2)}$. The same results also hold for the case $c\eta \geq 1, \lambda \geq 0$. Combining those facts with (53), we obtain

$$\mathbb{E}e^{\lambda(z+b)^2} \leq \sqrt{e^{2cn\eta(1+cn\eta)\lambda}}\sqrt{e^{2\lambda(2cn\eta b+b^2)}} \tag{54}$$

$$= e^{cn\eta(1+cn\eta)\lambda + 2cn\eta b\lambda + b^2\lambda} \tag{55}$$

completing the proof. $\square$

## B.3 MULTINOMIAL RANDOM VARIABLES

**Theorem B.5.** *Consider $\boldsymbol{z} = (z_1, \ldots, z_K)$ which follows the multinomial distribution with parameters $n$ and $(p_1, ..., p_K) > 0$, and $a_1, ..., a_K$ are non-negative functions which may depend on $\boldsymbol{z}$. Let $\boldsymbol{T} = \{j : z_j > 0\}$ and*

$$a_o = \begin{cases} 0 & \text{if } |\boldsymbol{T}| = K, \\ (\sum_{k \notin \boldsymbol{T}} p_k)^{-1} \sum_{k \notin \boldsymbol{T}} a_k p_k & \text{otherwise} \end{cases}$$

*For any constant $\delta > 0$, we have*

$$\Pr\left(\sum_{i=1}^K a_i(p_i - \frac{z_i}{n}) \geq (a_o + \sum_{j \in \boldsymbol{T}} a_j)\sqrt{\frac{-\ln\delta}{2n}}\right) \leq \delta \tag{56}$$

$$\Pr\left(\sum_{i=1}^K a_i(\frac{z_i}{n} - p_i) \geq (a_o + \sum_{j \in \boldsymbol{T}} a_j)\sqrt{\frac{-\ln\delta}{2n}}\right) \leq \delta \tag{57}$$

*Proof.* Denote $\bar{a} = \sum_{i \in \boldsymbol{T}} a_i$ and $i^* = \arg\max_{i \in \boldsymbol{T}}(p_i - \frac{z_i}{n})$. We first observe that

$$\sum_{i \in \boldsymbol{T}} a_i(p_i - \frac{z_i}{n}) = \bar{a}\sum_{i \in \boldsymbol{T}} \frac{a_i}{\bar{a}}(p_i - \frac{z_i}{n}) \tag{58}$$

$$\leq \bar{a}(p_{i^*} - \frac{z_{i^*}}{n}) \tag{59}$$

where the above inequality comes from the fact that $\sum_{i \in \boldsymbol{T}} \frac{a_i}{\bar{a}}(p_i - \frac{z_i}{n})$ is a convex combination of individual quantities $(p_i - \frac{z_i}{n})$ and hence is at most $p_{i^*} - \frac{z_{i^*}}{n}$.

Since $\boldsymbol{z}$ is a multinomial random variable, each element $z_{i^*}$ is a binomial random variable with parameters $n$ and $p_{i^*}$. Lemma B.2 shows the following for any $\epsilon > 0$:

$$\Pr\left(p_{i^*} - \frac{z_{i^*}}{n} \geq \epsilon\right) \leq e^{-2n\epsilon^2} \tag{60}$$

Combining this with (59) leads to the following:

$$\Pr\left(\sum_{i \in \boldsymbol{T}} a_i(p_i - \frac{z_i}{n}) \geq \bar{a}\epsilon\right) \quad \leq \quad \Pr\left(\bar{a}(p_{i^*} - \frac{z_{i^*}}{n}) \geq \bar{a}\epsilon\right) \tag{61}$$

$$\leq \quad \Pr\left(p_{i^*} - \frac{z_{i^*}}{n} \geq \epsilon\right) \tag{62}$$

$$\leq \quad e^{-2n\epsilon^2} \tag{63}$$

Choosing $\epsilon = \sqrt{\frac{-\ln\delta}{2n}}$ will lead to the first statement when $|\boldsymbol{T}| = K$.

Next we consider the cases of $|\boldsymbol{T}| < K$. Detnote $z_o = \sum_{k \notin \boldsymbol{T}} z_k$ and $\boldsymbol{T}' = \boldsymbol{T} \cup \{o\}$ and $p_o = \sum_{k \notin \boldsymbol{T}} p_k$. Using the same arguments before for the set $\boldsymbol{T}'$, we can show that the following holds with probability at least $1 - \delta$:

$$\sum_{i \in \boldsymbol{T}} a_i(p_i - \frac{z_i}{n}) + a_o(p_o - \frac{z_o}{n}) \leq \sum_{i \in \boldsymbol{T}} a_i\sqrt{\frac{-\ln\delta}{2n}} + a_o\sqrt{\frac{-\ln\delta}{2n}} \tag{64}$$

From the definition of $p_o$ and $a_o$, observe that $a_o\frac{z_o}{n} = 0 = \sum_{k \notin \boldsymbol{T}} a_k\frac{z_k}{n}$ and $a_o\sum_{k \notin \boldsymbol{T}} p_k = \sum_{k \notin \boldsymbol{T}} a_k p_k$. Plugging those terms into (64) will lead to the first result.

The second result can be proven by the same arguments, completing the proof. $\square$

**Lemma B.6** (Multinomial variable). *Consider a multinomial random variable* $(n_1, ..., n_K)$ *with parameters* $n$ *and* $(p_1, ..., p_K)$. *For any* $\delta > 0$:

$$\Pr\left(\sum_{i=1}^{K} p_i^2 \geq \sum_{i=1}^{K} \left(\frac{n_i}{n}\right)^2 + \sqrt{-\frac{2\ln\delta}{n}}\right) \quad \leq \quad \delta \tag{65}$$

$$\Pr\left(\sum_{i=1}^{K} \left(\frac{n_i}{n}\right)^2 \geq \sum_{i=1}^{K} p_i^2 + \sqrt{-\frac{2\ln\delta}{n}}\right) \quad \leq \quad \delta \tag{66}$$

*Proof.* Observe that

$$\sum_{i=1}^{K} p_i^2 - \sum_{i=1}^{K} \left(\frac{n_i}{n}\right)^2 \quad = \quad \sum_{i=1}^{K} \left[p_i^2 - \left(\frac{n_i}{n}\right)^2\right] \tag{67}$$

$$= \quad \sum_{i=1}^{K} \left[p_i + \frac{n_i}{n}\right]\left[p_i - \frac{n_i}{n}\right] \tag{68}$$

$$= \quad 2\sum_{i=1}^{K} \left(0.5p_i + \frac{0.5n_i}{n}\right)\left(p_i - \frac{n_i}{n}\right) \tag{69}$$

$$\leq \quad 2\max_{i \in [K]} \left(p_i - \frac{n_i}{n}\right) \tag{70}$$

where the last inequlality can be derived by using the fact that $\sum_{i=1}^{K} \left(0.5p_i + \frac{0.5n_i}{n}\right)\left(p_i - \frac{n_i}{n}\right)$ is a convex combination of the elements in $\{p_i - \frac{n_i}{n} : i \in [K]\}$, because of $1 = \sum_{i=1}^{K} \left(0.5p_i + \frac{0.5n_i}{n}\right)$. Furthermore, since $n_i$ is a binomial random variable with parameters $n$ and $p_i$, Lemma B.2

shows that $\Pr\left[p_i - \frac{z_i}{n} \geq \epsilon\right] \leq e^{-2n\epsilon^2}$, for any $\epsilon \geq 0$. By choosing $\epsilon = \sqrt{\frac{-\ln\delta}{2n}}$, this immediately implies $\Pr\left(p_i - \frac{n_i}{n} \geq \sqrt{\frac{-\ln\delta}{2n}}\right) \leq \delta$. Combining this fact with (70), we obtain $\Pr\left(\sum_{i=1}^{K} p_i^2 - \sum_{i=1}^{K}\left(\frac{n_i}{n}\right)^2 \geq 2\sqrt{\frac{-\ln\delta}{2n}}\right) \leq \delta$. By using the same arguments, one can show the second result, completing the proof. $\qquad\square$

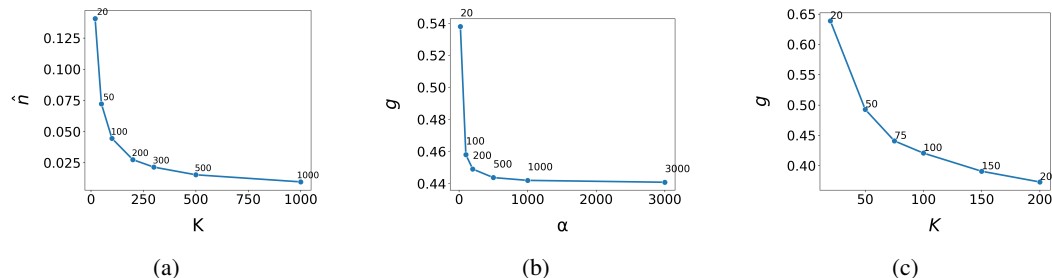

(a)    (b)    (c)

Figure 3: (a) The term $\hat{n} = \sum_{i=1}^{K} \left(\frac{n_i}{n}\right)^2$ as $K$ changes. As expected, as $K$ becomes larger, the sum of the squared ratios becomes smaller. (b) reports the uncertainty $g = C\sqrt{\hat{u}\alpha \ln \gamma} + C\sqrt{\frac{\ln(2/\delta)}{2n}}$ as $\alpha$ changes (for fixed $K = 75$) and (c) as $K$ changes (for fixed $\alpha = 3000$). Those quantities were computed on the COCO dataset.

## C    MORE EXPERIMENTAL SETUPS AND RESULTS

This section provides more details about our experiments for both problems and more results. Some ablation studies are also reported to see the different behaviors of some parameters on our bounds.

### C.1    COMMON PREPROCESSING AND SETTINGS

*Data preprocessing and space partition:*

- We first preprocessed the images following PyTorch[2]:  The images are resized to $resize\_size = [256]$ using interpolation=InterpolationMode.BILINEAR, followed by a central crop of $crop\_size = [224]$. Finally the values are first rescaled to $[0.0, 1.0]$. Those operations are required for Pytorch pretrained models.
- To build a partition $\Gamma$ with any give $K$ small areas, we randomly choose $K$ points in $[0.0, 1.0]^{C \times H \times W}$ to be the centroids, since each preprocessed image belongs to $[0.0, 1.0]^{C \times H \times W}$. Those centroids are used to build the small areas $\mathcal{Z}_i$ in the partition.
- Each training image $x$ will be assigned to area $\mathcal{Z}_i$ if it is closest to the centroid of $\mathcal{Z}_i$ amongst all centroids, according to the Euclidean distance. By this way, we can determine $S_i$ (or $D_i$) when given $S$ (or $D$).

### C.2    EFFECT OF SOM PARAMETERS

Note that our bound depends on the choice of some parameters. Figure 3 reports the changes of $\sum_{i=1}^{K} \left(\frac{n_i}{n}\right)^2$ as the partition $\Gamma$ changes. We can see that this quantity tends to decrease as we divide the input space into more small areas. Meanwhile, Figure 3 reports the uncertainty term, as either $\alpha$ or $K$ changes. Observe that a larger $K$ can increase the uncertainty fast, while an increase in $\alpha$ can gradually decrease the uncertainty. Those figures enable an easy choice for the parameters in our bound.

### C.3    CORRELATION WITH THE DYNAMICS OF TEST ERROR OVER THE TRAINING COURSE: MORE RESULTS

In Section 5.2, we mention training a ResNet18 architecture from scratch on a 20,000 sample set from CIFAR10, while 10,000 is held out for validation. We then compute the macro-average error $mac_h$ and the test error (on a different test set) on each epoch. The training is done in three separate settings, each varying in details such as learning rate and scheduler to induce varying levels of capacity.

---

[2]https://pytorch.org/vision/0.20/models/generated/torchvision.models.vit_b_16.html

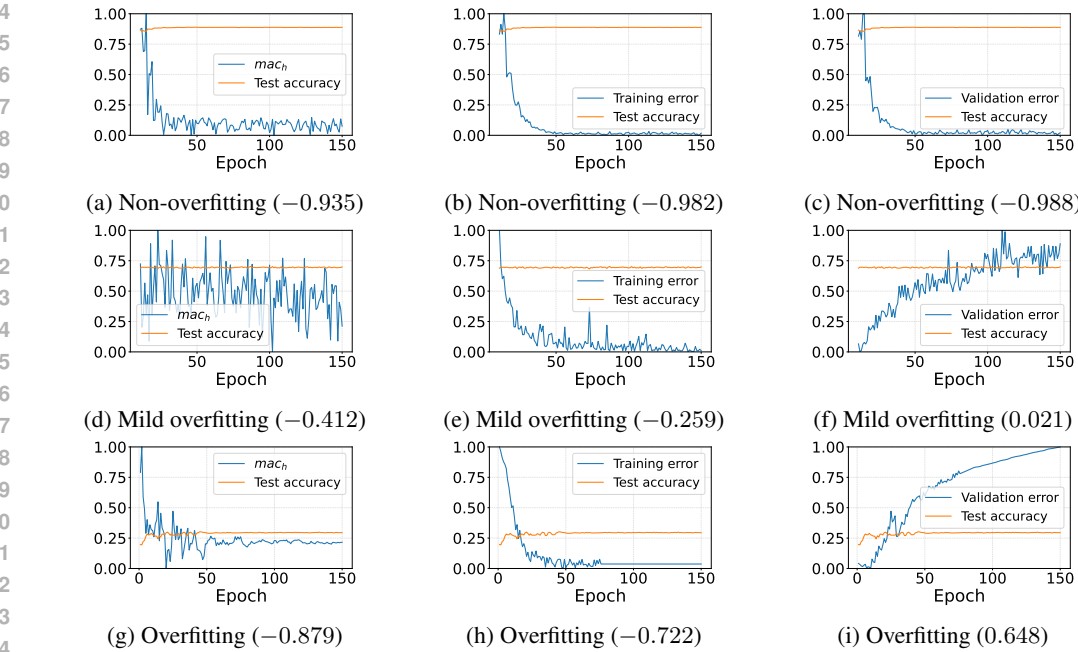

(a) Non-overfitting ($-0.935$)  (b) Non-overfitting ($-0.982$)  (c) Non-overfitting ($-0.988$)

(d) Mild overfitting ($-0.412$)  (e) Mild overfitting ($-0.259$)  (f) Mild overfitting ($0.021$)

(g) Overfitting ($-0.879$)  (h) Overfitting ($-0.722$)  (i) Overfitting ($0.648$)

Figure 4: Correlation of $mac_h$, training error, and validation error (respectively) to test accuracy during training **ResNet18** on **FashionMNIST**. Each row corresponds to a different trained model. The correlation coefficient is shown in brackets - a lower correlation value is better.

We provide additional results on a different model (the DenseNet121 architecture) and a dataset (FashionMNIST). The results are reported by Figures 4, 5, and 6. We can observe a common pattern from those figures. Across models and datasets, throughout epochs in the training process, $mac_h$ showcases a remarkable capacity to correlate with true model capibilties. This is even more noticeable in overfitting settings, where the simplistic training or validation error demonstrate a significantly lowered correlation, while $mac_h$ feigns much better.

Those figures provide a consistent behavior of $mac_h$: (1) for well-trained models, $mac_h$ is often small and stabilizes in the ending epochs; (2) for mildly-overfitted models, $mac_h$ is often unstale and high although the training error stabilizes; (3) for highly-overfitted models, $mac_h$ is both high and stable in the ending epochs. Those behaviors suggest that $mac_h$ can be an easy and effective tool to track the true error of a model.

### C.4 THE ROLE OF MACRO-LEVEL BEHAVIORS

We continue investigating the capacity of macro-level component in Bound (4) across 32 large-scale NNs. Table 3 displays the result, when computing the main terms in our bound from either the training set validation set. We observe that there is a strong correlation between macro-level behavior (hidden in the second term of our bound) and the true error of the models. This result combined with the previous one demonstrates the significance of the macro-level behavior when assenssing a model.

### C.5 ALIGNMENT BETWEEN THE PARTITION AND DATA GEOMETRY

*Is there any effect of the alignment between the partition and data geometry on our bounds?* This is an important question to evaluate our bounds. Since it would be challenging to see a good alignment for high-dimensional settings, we opt to a controlled simulation so that one can easily see when a partition aligns well with the data distribution.

**Data distribution:** we use a 2-dimensional *Gaussian mixture model* (GMM) with $K = 100$ components and variance $\nu$. In this GMM, each sample $(x, y)$ is generated by the following process:

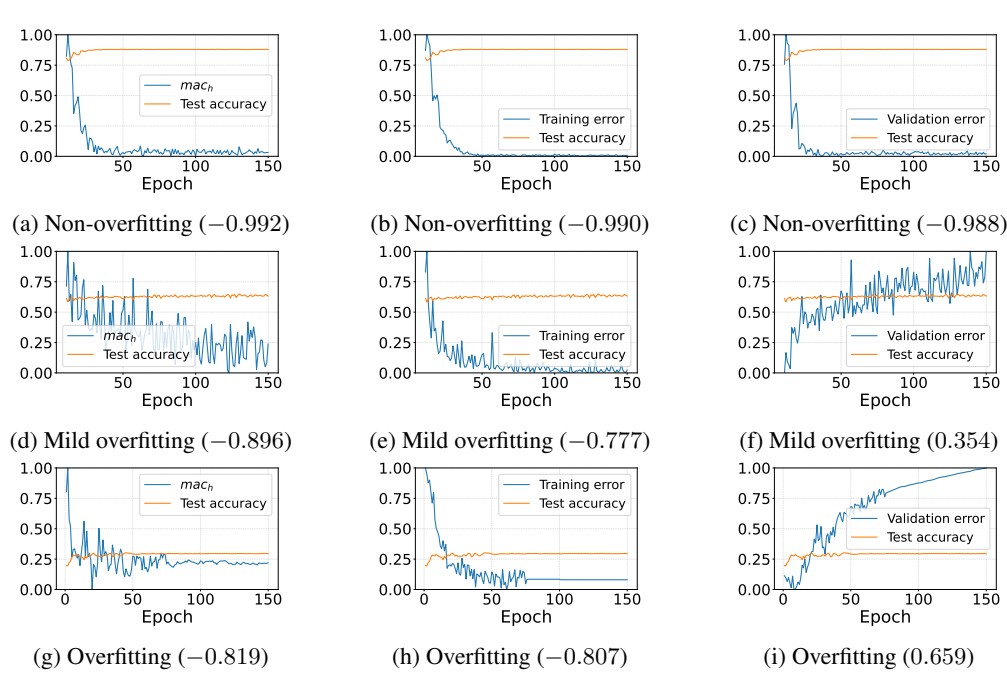

Figure 5: Correlation of $mac_h$, training error, and validation error (respectively) to test accuracy during training **DenseNet121** on **CIFAR10**. Each row corresponds to a different trained model. The correlation coefficient is shown in brackets - a lower correlation value is better.

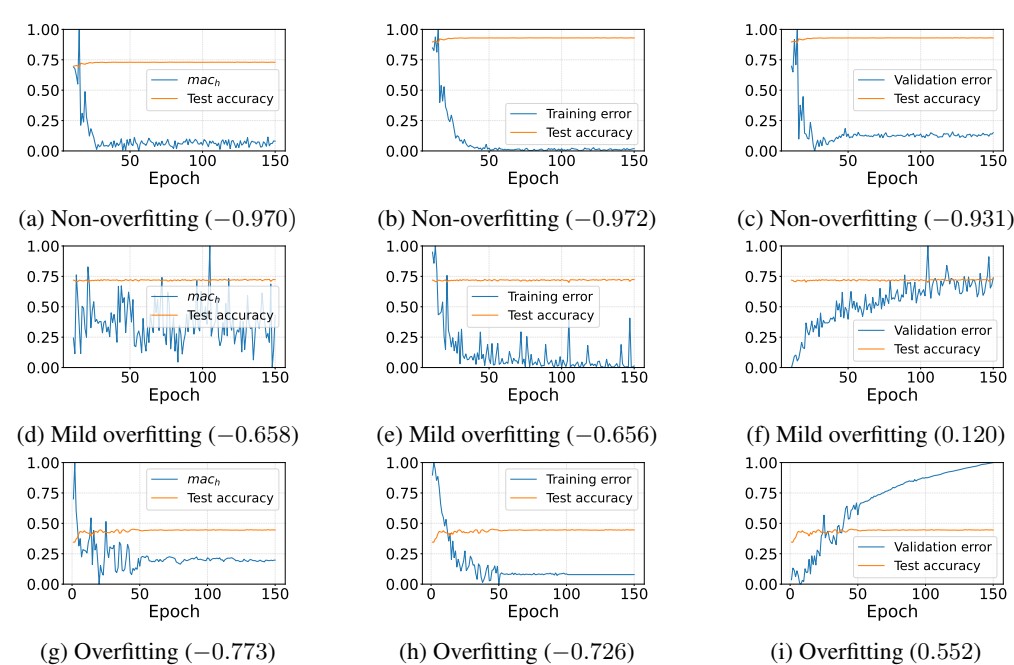

Figure 6: Correlation of $mac_h$, training error, and validation error (respectively) to test accuracy during training **DenseNet121** on **FashionMNIST**. Each row corresponds to a different trained model. The correlation coefficient is shown in brackets - a lower correlation value is better.

Table 3: Correlation of the main components in Bound (4) to Test error on PyTorch's pre-trained classification models. Those components are computed from either the ImageNet training set or validation set. Here $A2 = \frac{b}{n}mac_h$ denotes the second term in Bound (4).

| Model | Test error | S = Train | | S = Val | |
|---|---|---|---|---|---|
| | | Train error | A2 | Val error | A2 |
| ResNet18 V1 | 0.3031 | 0.2124 | 0.0615 | 0.3024 | 0.8035 |
| ResNet34 V1 | 0.2675 | 0.1567 | 0.0457 | 0.2670 | 0.7248 |
| ResNet50 V1 | 0.2388 | 0.1312 | 0.0382 | 0.2385 | 0.6482 |
| ResNet101 V1 | 0.2268 | 0.1050 | 0.0305 | 0.2263 | 0.6053 |
| ResNet152 V1 | 0.2178 | 0.1013 | 0.0293 | 0.2169 | 0.6084 |
| ResNet50 V2 | 0.1918 | 0.8936 | 0.0252 | 0.1915 | 0.5195 |
| ResNet101 V2 | 0.1819 | 0.6008 | 0.0171 | 0.1810 | 0.4941 |
| ResNet152 V2 | 0.1777 | 0.5178 | 0.0148 | 0.1771 | 0.4739 |
| SwinTransformer B | 0.1652 | 0.6464 | 0.0188 | 0.1642 | 0.4402 |
| SwinTransformer B V2 | 0.1596 | 0.6392 | 0.0183 | 0.1589 | 0.4346 |
| SwinTransformer T | 0.1859 | 0.9992 | 0.0287 | 0.1853 | 0.5037 |
| SwinTransformer T V2 | 0.1802 | 0.8724 | 0.0254 | 0.1793 | 0.4883 |
| VGG13 | 0.3011 | 0.1846 | 0.0527 | 0.3006 | 0.8124 |
| VGG13 BN | 0.2853 | 0.1922 | 0.0557 | 0.2845 | 0.7689 |
| VGG19 | 0.2770 | 0.1612 | 0.0461 | 0.2761 | 0.7336 |
| VGG19 BN | 0.2576 | 0.1594 | 0.0464 | 0.2576 | 0.6997 |
| DenseNet121 | 0.2563 | 0.1563 | 0.0454 | 0.2556 | 0.7016 |
| DenseNet161 | 0.2296 | 0.1048 | 0.0304 | 0.2289 | 0.6129 |
| DenseNet169 | 0.2436 | 0.1240 | 0.0357 | 0.2441 | 0.6784 |
| DenseNet201 | 0.2310 | 0.9806 | 0.0287 | 0.2312 | 0.6486 |
| ConvNeXt Base | 0.1607 | 0.5209 | 0.0149 | 0.1594 | 0.4255 |
| ConvNeXt Large | 0.1563 | 0.3846 | 0.0108 | 0.1558 | 0.4188 |
| RegNet Y 128GF e2e | 0.1178 | 0.5565 | 0.0161 | 0.1178 | 0.3190 |
| RegNet Y 128GF linear | 0.1396 | 0.9032 | 0.0266 | 0.1393 | 0.3657 |
| RegNet Y 32GF e2e | 0.1322 | 0.7127 | 0.0204 | 0.1315 | 0.3531 |
| RegNet Y 32GF linear | 0.1548 | 0.1056 | 0.0310 | 0.1538 | 0.4105 |
| RegNet Y 32GF V2 | 0.1678 | 0.3761 | 0.0107 | 0.1663 | 0.4431 |
| ViT B 16 linear | 0.1805 | 0.1497 | 0.0438 | 0.1811 | 0.4935 |
| ViT B 16 V1 | 0.1895 | 0.5916 | 0.0168 | 0.1893 | 0.4957 |
| ViT H 14 linear | 0.1434 | 0.9951 | 0.0289 | 0.1429 | 0.3891 |
| ViT L 16 linear | 0.1488 | 0.1100 | 0.0324 | 0.1486 | 0.4110 |
| ViT L 16 V1 | 0.2040 | 0.3465 | 0.0097 | 0.2034 | 0.5546 |
| **Correlation to Test error** | | 0.7965 | 0.7892 | 0.9999 | 0.9976 |

- Draw an index $z \sim \text{Cat}(1/K, \ldots, 1/K)$,
- Generate $x \sim \mathcal{N}(\mu_z, \nu)$, where $\mu_z = (\pi z, 0) \in \mathbb{R}^2$,
- The class label is set to $y = 1$ if $z$ is odd and $y = 0$ otherwise.

**Partitioning strategies:**

- **T1** uses a uniform grid that divides the data space into $K$ equally sized regions. Though simple, this may misalign with the true data distribution.
- **T2** generates $K$ centroids uniformly at random to form the partition, which may still lead to misalignment.
- **T3** fixes the centroids as mixture means $\mu_1, \ldots, \mu_K$ to create regions with balanced probabilities (i.e., $p_i \approx p_j, \forall i, j$).

**Baselines:** We compare bound (4) with the one by Than & Phan (2025), since they both exhibit a strong dependence on the data-partition alignment. Ignoring the empirical error, we focus on the uncertainty parts in those computable bounds:

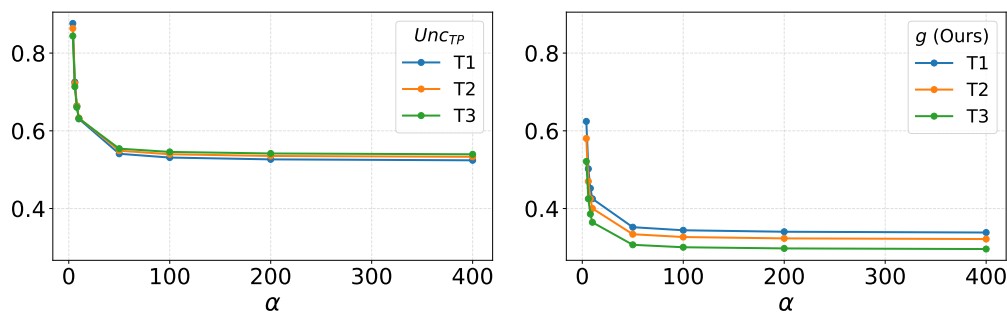

Figure 7: The role of a good alignment between the partition and data geometry. Three different partitions are investigated. This figure depicts the dynamics of the uncertainty part ($g$ and $Unc_{TP}$) when changing $\alpha$.

$$g = \frac{b}{n} \sum_{i \in \boldsymbol{T}} F(\boldsymbol{S}_i, \boldsymbol{h}) + C\sqrt{\hat{u}\alpha \ln \gamma} + C\sqrt{\frac{\ln(2/\delta)}{2n}} \qquad \text{(Ours)}$$

$$Unc_{TP} = C\sqrt{\hat{a}\alpha \ln \gamma} + \frac{C(1+\sqrt{2})\sqrt{\ln(4K/\delta)}}{n} \sum_{i \in \boldsymbol{T}} \sqrt{n_i} + \frac{4C|\boldsymbol{T}|\ln(4K/\delta)}{n} \qquad \text{(Than \& Phan, 2025)}$$

where $\hat{a} = \frac{\gamma}{2n} + \frac{\gamma^2}{2} \sum_{i \in \boldsymbol{T}} \left(\frac{n_i}{n}\right)^2 + \gamma^2 \sqrt{\frac{2}{n} \ln \frac{2K}{\delta}}$.

We sample 100000 data points from the GMM before with variance $\nu = 1$. Figure 1 provides a visualization for 4 components. This data set is used to train a simple MLP (obtaining an accuracy of 88.44%), which is then used to compute $g$ and $Unc_{TP}$.

Figure 7 reports the results, for varying values of $\alpha$. One can observe that **T1** often produces the worst result, while **T3** consistently produces the best results. Note that among three parition strategies, **T3** seems to best balance the local probabilities, and hence can produce smallest uncertainty. The good geometrical alignment of T3 results in a lower $g$ and $Unc_{TP}$.

Observe that $g$ is often much smaller than $Unc_{TP}$ for the same setting. Even for the best scenario, $Unc_{TP}$ is much worse than $g$ in our bound. This demonstrates the benefits of our novel concentration in Section 4.

### C.6 CONCENTRATION COMPARISON: MORE RESULTS

Figure 8 reports the empirical values of $gap_O$ and $gap_K$, whose the coefficients are computed from 32 large-scale models, showcasing a realistic scenario. It shows that when most of the coefficients are small, our gap can be significantly tighter than the baseline. Figure 9 reports the two gaps when the size $K$ (number of variables) increases. The result also exhibits the same pattern. Our gap can be significantly better for more accurate models.

### C.7 TRADITIONAL ML MODELS

In addition to deep learning models on image data, we also evaluate the behavior of the proposed generalization bound (4) on the tabular *Diabetes 130* (3-class) and *News Aggregator* (4-class) datasets[3], using traditional machine learning approaches: Linear SVM, Bernoulli Naive Bayes, and LightGBM. Each model was trained (after applying the standard preprocessing, i.e., one-hot encoding of categorical features and vectorizing text data via TF-IDF) on the full training split. Afterwards, to evaluate the bound, we generated $K = 100$ clusters over the training set, similar to how we approached image classification, as mentioned in Section 5. Across five seeds, we produced five bound values per model, from which we report the mean and standard deviation.

---

[3]Datasets available at the UCI Repository

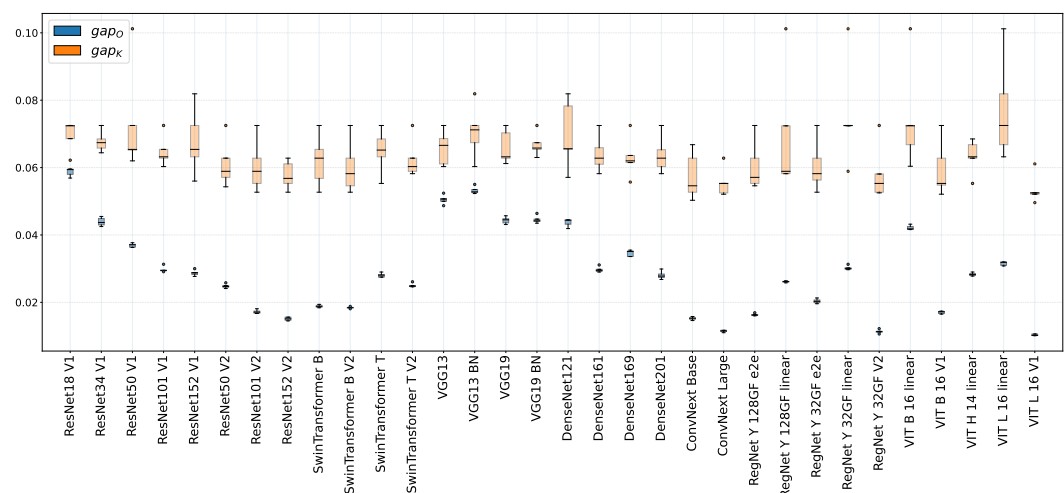

Figure 8: Comparison of the empirical values of $gap_O$ and $gap_K$ across 32 pretrained PyTorch classification models with $K = 200$.

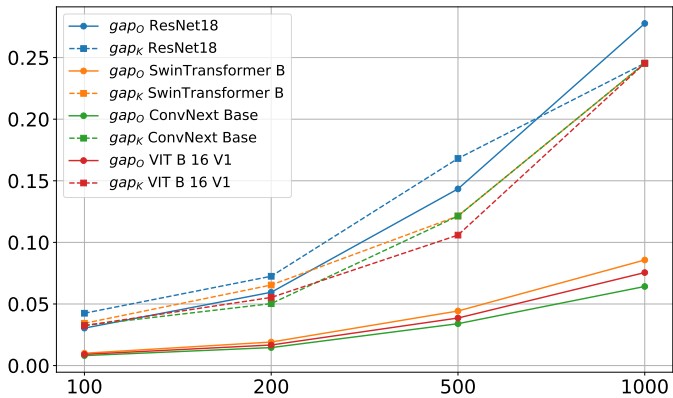

Figure 9: Comparison of $gap_O$ (solid lines) and $gap_K$ (dashed lines) across different classifiers and varying number of clusters $K$. Results are averaged over five runs with random seeds.

Table 4: Error estimates on traditional machine learning models, for two datasets. Each row show results for a model. All values are in %.

(a) Results on the tabular *Diabetes 130* dataset.

| Model | Train error | Test error | Error bound | |
|---|---|---|---|---|
| | | | Than & Phan (2025) | Ours (4) |
| Linear SVM | 39.68 | 41.97 | $98.84 \pm 0.05$ | $88.10 \pm 0.19$ |
| Bernoulli Naive Bayes | 42.15 | 43.88 | $101.32 \pm 0.05$ | $91.98 \pm 0.22$ |
| LightGBM | 35.77 | 40.29 | $94.93 \pm 0.05$ | $81.69 \pm 0.20$ |

(b) Results on the tabular *News Aggregator* dataset.

| Model | Train error | Test error | Error bound | |
|---|---|---|---|---|
| | | | Than & Phan (2025) | Ours (4) |
| Linear SVM | 3.06 | 5.27 | $53.04 \pm 0.88$ | $40.26 \pm 0.86$ |
| Multinomial Naive Bayes | 6.56 | 7.50 | $56.54 \pm 0.88$ | $44.63 \pm 0.84$ |
| LightGBM | 8.20 | 8.77 | $58.18 \pm 0.88$ | $46.38 \pm 0.85$ |

The results are reported in Table 4. Across all models, though the bound was consistently larger than the observed test error, the correlation between the bound and test error is high, with relatively small variance across seeds, demonstrating that the bound behaves coherently and is sensitive to, thus can

be indicative of, differences in model quality. For example, LightGBM and Linear SVM achieving the lowest bounds among the three models for Diabetes 130 and News Aggregator, respectively, corresponding to test errors. In addition, our results consistently outperform a prior work, thus highlighting a tighter bound.

## D  RATE OF CONVERGENCE

Bound (1) in Theorem 3.1 contains three main contributions:

- the empirical term $F(S, h)$,
- the macro term $\dfrac{b}{n} \sum_{i \in T} F(S_i, h)$, where $b = \sqrt{0.5n \ln(1/\delta_1)}$ so $\dfrac{b}{n} = O(1/\sqrt{n})$ and thus the macro term decays like $O(n^{-1/2})$ (up to log factors and the local errors),
- the uncertainty term $C\sqrt{\dfrac{u \ln(1/\delta_2)}{2n^2}}$, where $u = \gamma n(1 + 2b) + |T|b^2 + \gamma^2 n^2 \sum_i p_i^2$.

The dominant contribution $u$ depends on the partition and data distribution:

- if $\sum_i p_i^2 = O(1/K)$ and $K$ grows sublinearly, the $\gamma^2 n^2 \sum_i p_i^2$ piece is $O(n^2/K)$, so $\sqrt{u/n^2} = O(K^{-0.5})$ and the uncertainty term is $O(K^{-0.5})$;
- in balanced partitions with $K = O(n^\beta)$ for $\beta \leq 1$, the uncertainty term is $O(n^{-\beta/2})$.

In summary, for balanced partitions with $K = O(n^\beta)$ for $\beta \leq 1$, the bound is

$$F(S, h) + O(n^{-1/2}) \sum_{i \in T} F(S_i, h) + O(n^{-\beta/2}) + O(n^{-1/2})$$

