# OpenReview forum: "A Tight  Error Bound for Deep Learning via Distribution and Loss Complexity"
_ICLR.cc/2026/Conference — Submitted to ICLR 2026_

### Official Review · Reviewer_K75M · 2025-10-26

**Soundness:** 3
**Presentation:** 3
**Contribution:** 3
**Rating:** 6
**Confidence:** 3

**Summary:**

This paper tackles the long-standing problem of non-vacuous generalization bounds for modern deep nets. It proposes a model-dependent bound that explicitly captures both data-distribution complexity and loss-landscape structure via a partition of the input space, emphasizing how alignment between data geometry and the loss landscape drives generalization. A key technical contribution is a new concentration inequality for multinomial variables that removes the usual $\sqrt{\ln K}$ dependence, yielding tighter rates and enabling a sharper bound. Building on this, the authors derive a computable bound—estimable from training data under i.i.d. and bounded-loss assumptions—that preserves the theory’s structure and avoids the looseness of prior results (notably improving over Than & Phan, 2025). Empirically, across 37 pretrained models (32 ImageNet classifiers, 5 COCO segmenters), the bound delivers the tightest non-vacuous error estimates among compared methods (e.g., ResNet-18: ~57.9% → ~54.7% vs. ~30.3% true error). Moreover, a macro-level term in the bound correlates well with test performance and can track generalization dynamics—even when validation error is misleading—suggesting value as a diagnostic tool for overfitting and model selection. Overall, the paper presents a well-motivated, technically solid advance that meaningfully tightens practical generalization guarantees for deep learning.

**Strengths:**

- Data/Loss-Aware Bounds, Sharper Concentration, and a Computable Guarantee: The paper introduces a model-dependent generalization bound that captures micro-level (local regions) and macro-level (across the data manifold) behavior via a data-space partition, revealing a previously uncharacterized interaction between data geometry and the loss landscape. A key advance is a new concentration inequality for multinomial variables that removes the usual $\sqrt{\ln K}$ dependence, yielding tighter rates and addressing a weakness in recent SOTA bounds (e.g., Than & Phan, 2025). The theory is presented as two complementary results: a general bound and an exactly computable bound that preserves the structure while requiring only i.i.d. sampling and a bounded loss. Empirically, the computable bound consistently outperforms prior SOTA, reflecting richer loss-landscape information and stronger distribution awareness.

 - Extensive Empirical Validation and Diagnostic Utility: Tested on 37 large-scale models (32 ImageNet classifiers, 5 COCO segmenters), the bound delivers consistently tighter non-vacuous estimates than prior methods across the board. A macro-level term from the theory also tracks test error during training, sometimes outperforming validation error as an indicator—suggesting practical use for model selection and overfitting detection beyond its theoretical guarantees.

**Weaknesses:**

- Complexity of the Bound and Partition Dependence: A potential weakness is the added complexity introduced by the bound’s dependence on a chosen data space partition (Γ). While the partition-based approach is key to capturing data geometry, it also means the bound comes with several terms (sums over partition regions, local losses $F(S_i,h)$, etc.) that may be hard to interpret intuitively. The paper does not fully discuss how to choose an optimal or canonical partition in practice – the tightness of the bound could depend on this choice. If $\Gamma$ is chosen poorly (e.g. misaligned with the data’s true structure), the bound might become looser. Thus, more guidance on selecting or learning a good partition (and the sensitivity to the partition size $K$) would strengthen the practicality of the approach. As it stands, applying the bound requires an additional design decision (how to partition the input space) that practitioners must make without a clear prescription from the paper.

 - Residual Gap to True Error: Despite the improvements, the bounds can still be somewhat loose in absolute terms for certain models. The authors report non-vacuous and tighter bounds than before, but there remains a noticeable gap between the bound values and the actual test errors in many cases. For example, as noted above, the ResNet-18 classifier with ~30% test error has a bound around ~54.7%, and other large networks (e.g. ResNet-152 V2) with ~17.8% test error have bounds in the low 30% range. While these are the tightest known results to date, they are still far from truly tight estimates of generalization performance. The paper could discuss the sources of this remaining gap, for example, whether it is due to unavoidable worst-case terms (e.g., union bounds or concentration slack), suboptimal partitioning, or other conservative aspects of the theory. A deeper analysis of what limits the bound’s tightness would help in understanding how much further this line of work can go.

 - Scope of Empirical Evaluation: The experiments, albeit extensive for vision models, are focused on image classification and segmentation tasks. It is not entirely clear how well the proposed bounds would extend to other domains such as natural language processing or other data modalities. The underlying theory is general, but certain elements (like how to partition text or sequence spaces, or handling very high-dimensional input differently) might pose new challenges. Similarly, all reported models are pre-trained and fixed; an interesting scenario is using the bound during training to monitor generalization. While the authors hint at tracking test error dynamics, they do not explicitly demonstrate using the bound online for early stopping or model selection.

 - Assumptions: The theoretical results assume an i.i.d. sample and a bounded loss function. These assumptions are standard for classification error (0-1 loss is bounded in $[0,1]$) and were likely satisfied in experiments (where error percentage is the metric). However, for settings using unbounded loss functions, it’s unclear if the bound is directly applicable or if one must switch to a bounded proxy. An explanation on the theoretical generalization bound results with the example of cross-entropy loss or mean-squared loss which are unbounded would give readers a better understanding.

**Questions:**

1. Could the authors elaborate on how the partition $\Gamma(Z)$ should be chosen in practice? The theory emphasizes that a well-aligned partition (reflecting data geometry and loss) is crucial for a tight bound. Did the experiments use a specific strategy (e.g., clustering data, using class labels, or random partitions) to define $\Gamma$? How sensitive is the bound’s tightness to the choice of partition and the number of regions $K$? Any guidance on selecting or tuning the partition for a new dataset would be very helpful for practitioners.

2. What do the authors see as the main factors contributing to the gap between the bound and the true test error in the empirical results? For instance, is the remaining slack mostly due to worst-case concentration terms (e.g., the $\ln(1/\delta)$ terms or the use of union bounds) or due to suboptimal alignment/partitioning in practice? Insight into whether further tightening is possible (perhaps by refining the bound or using data-dependent $\Gamma$ choices) would clarify the significance of the remaining gap and potential future improvements.

3. The theoretical development assumes a bounded loss function $0 \le \ell(h,z) \le C$. In the experiments, it appears the 0-1 classification error (and analogous segmentation error) was used to satisfy this. How would the framework handle common unbounded losses like cross-entropy or mean-squared error? Would one need to enforce a bound (e.g., clip losses or use a surrogate) to apply the theorem, or is there an extension of the concentration inequality that could accommodate sub-Gaussian or heavy-tailed losses? Clarification on this point would delineate the practical scope of the bounds.

4. The paper demonstrates the bound on vision tasks. Do the authors anticipate any challenges in applying it to other domains such as NLP or speech, where the input dimension and structure differ significantly? Also, given the bound’s ability to track generalization, have the authors considered using it for model selection or early stopping in lieu of a validation set? For example, could one compute the bound during training (on the training data) to decide when a model is generalizing well or beginning to overfit? Exploring such an application would highlight the bound’s utility in practice and we are curious if the authors have preliminary thoughts or results in this direction.

---

> ### Author Response · Authors · 2025-11-21
>
> We thank the reviewer for the constructive and insightful comments. Below, we address the main questions first, followed by the points raised under weaknesses.
>
> ### 1. The questions
> > **Q1.** How should the partition Γ be chosen in practice? What strategy was used? How sensitive is the bound?
>
> The paper already describes how $\Gamma$ is constructed in all experiments; we now clarify this explicitly.
>
> In all large-scale ImageNet and COCO experiments, the partition $\Gamma$ is built in a simple, fully unsupervised manner: a *random centroid-based partition* (No clustering, no class labels, and no model-dependent information was used). The details appear in Appendix C.1.
>
> *In the synthetic GMM experiments*, we additionally show three explicit strategies (T1–T3) purely for diagnostic purposes:  T1--uniform grid; T2--random centroids; T3--centroids placed at the GMM mixture means, yielding balanced region probabilities $p_i$  (Appendix C.5). These experiments illustrate the effects of good vs. bad alignment.
>
> **Sensitivity of the bound to $\Gamma$**: The paper provides direct quantitative evidence about
> -   The term $\sum_i (n_i/n)^2$ decreases as $K$ grows, showing its dependence on $\Gamma$.  (Figure 3a, Appendix C.2 )
> -   The uncertainty term in Theorem 3.2 varies predictably as $K$ or $\alpha$ changes (Figure 3b–c).
> -   In the synthetic GMM diagnostic, T3—which best balances $p_i$—gives the smallest uncertainty; T1 (severely misaligned) yields the worst.
>
> Thus, sensitivity is aligned with the theory: partitions that equalize local probabilities reduce $\sum_i p_i^2$, improving the bound. For large $\alpha$, our bound is surprisingly stable, suggesting an ease of choosing partition.
>
> **Guidance for practitioners**: Our findings suggest that
>
> -   *The simple random-centroid partition already works well in high-dimensional image spaces.* Despite its simplicity, it produces the tightest known computable bounds for 37 large-scale models.
> -   *Increasing K produces a predictable trade-off* (Figure 3): lower $\sum_i (n_i/n)^2$ but potentially larger $|T|$, affecting the macro term $mac_h$. Users can pick $K$ by examining these two terms, as discussed in Appendix C.2.
> -   A suitable partition may result in a tighter bound. But a large value of $\alpha$ can ease this step.
>
> We will make the above experimental details more explicit in the paper.
>
> >**Q2.** What drives the remaining gap between the bound and true test error?
>
> We observe two main factors arisen directly from the bound structure:
>
> - **(1) Unavoidable concentration slack**: The uncertainty terms in both Theorem 3.1 and Theorem 3.2 contain expressions that reflect worst-case fluctuations of multinomial counts, e.g.,  $C \sqrt{u\ln(1/\delta_2)/(2n^2)}$.  Even with the new concentration inequality (Theorem 4.1), which removes the usual $\sqrt{\ln K}$ scaling of prior work, some conservative slack remains due to the need to upper-bound unknown quantities ($p_i$ and $a_i(h)$) and the union over regions. This is intrinsic to any distribution-free, worst-case concentration.
>
> - **(2) Imperfect data-partition alignment in practice**: We do not use the optimal partition; ImageNet/COCO experiments rely solely on *random centroids*. Given high dimensionality, this already provides good empirical performance, but not the optimal alignment suggested by the theory. Our synthetic experiments illustrate this clearly: T3 (aligned) ≪ T2 (random) ≪ T1 (uniform grid), in terms of uncertainty (Figure 7). Thus, further tightening may come from future work.
>
> - **(3) Real models have nonzero macro-level errors in several regions**: Even state-of-the-art models exhibit pockets of high local error that inflate $mac_h$. This is not a weakness of the theory but an actual _diagnostic_: the macro term correlates well with test error across models (Table 3).
>
> > **Q3.** How does the framework handle unbounded losses (e.g., cross-entropy, MSE)?
>
> For unbounded losses such as cross-entropy or MSE where $C=\infty$, our bounds may not directly work. One would need to either:
>   - use a bounded surrogate (e.g., clipped loss), or
>   - use smoothing techniques (as used in [1, 2]) to make the bounds applicable.
>   - adapt the concentration step to handle sub-Gaussian or sub-exponential tails.
>
> [1] Lofti et al. Non-vacuous generalization bounds for large language models. In ICML, 2024.
>
> [2] Lofti et al. Unlocking tokens as data points for generalization bounds on larger language models. In NeurIPS, 2024.

---

> > ### Author Response · Authors · 2025-11-21
> > **Response to Reviewer K75M (continue)**
> >
> > > **Q4.** Applicability beyond vision, and using the bound during training
> >
> > **General applicability**:
> > The theory itself is *domain-agnostic*. It takes any measurable space $Z$, any partition $\Gamma$ of that space, and requires only i.i.d. sampling and a bounded loss. The only domain-specific ingredient is the **design of $\Gamma$**. For images, random Euclidean centroids are natural because preprocessing maps inputs into a normalized cube.
> >
> > For NLP or speech, one should ensure the i.i.d. assumption. If it is assured, then
> > -   one could embed tokens/sequences into a vector space using pretrained embeddings,
> > -   or use Hamming- or edit-distance-based Voronoi regions,
> > -   or use model representations (e.g., hidden states) as the metric space.
> >
> > The theory remains unchanged; only the partition instantiation differs.
> >
> > We did a set of experiments with tabular (Diabetes) and a text (News Aggregator) datasets, and measured our bound for some traditional ML models. The results have been added in Appendix C.7 of the revised manuscript.
> >
> > **Model selection or early stopping**:
> > The paper already contains a **preliminary exploration** of the bound’s diagnostic power. Section 5.2 (and Appendix C.3) shows that the macro term $mac_h$ tracks test error dynamics across training epochs, sometimes better than validation error.
> >
> > Using the bound itself (e.g., the computable term in Theorem 3.2) during training is feasible—since all components are computable from the training set—but we focused the paper on post-hoc evaluation. Extending the dynamic experiments to use the full bound is straightforward and will be discussed as a direction for future work.
> >
> > ### 2. Other concerns
> >
> > > **C1.** “Complexity of the bound and partition dependence”
> >
> > The dependence on $\Gamma$ is intentional: our bounds capture a data-geometry–loss-landscape interaction absent from prior theory. The paper already provides:
> > -   qualitative illustrations (Figure 1) highlighting alignment and misalignment,
> > -   quantitative diagnostics (Figure 3) showing how key terms evolve with $K$,
> > -   synthetic experiments with three partitioning strategies (Appendix C.5).
> >
> > For large-scale datasets, we deliberately adopted the simplest possible choice—random centroids—precisely to avoid heavy hyperparameter tuning. Despite this simplicity, the bound significantly outperforms prior work.
> >
> > We agree that adding more explicit guidance in the main text would enhance clarity. Since it would be challenging to see a good alignment for high-dimensional settings, our simulation in Appendix C.5, ablation in Appendix C.2 and the goodness of a random partition for large-scale datasets can suggest some practical guidances to use our bound.
> >
> > > **C2.** “Residual gap to true error”
> >
> > As detailed in Q2 before, the remaining gap is primarily due to: unavoidable worst-case concentration slack, the fact that $\Gamma$ is not aligned on real datasets, genuine model imperfections captured by the macro-level term. These are intrinsic challenges of partition-based generalization bounds. Importantly, our method already eliminates the $\sqrt{\ln K}$ dependence that limited prior work and achieves the tightest computable bounds known so far.
> >
> > We will expand the discussion of what limits further tightening and why the remaining gap is structurally expected.
> >
> > > **C3.** “Scope of empirical evaluation”
> >
> > The paper aims to provide the **first evidence** that a bound explicitly encoding micro- and macro-level behaviors can scale to dozens of ImageNet-level models. Extending to other domains is conceptually straightforward, as explained in Q4 before.
> >
> > Regarding monitoring: our training experiments for CIFAR-10 and FashionMNIST already reveal that the macro term tracks generalization extremely well. Extending this to using the full bound during training is a promising direction and will be highlighted in the revised discussion.

---

### Official Review · Reviewer_bN2N · 2025-10-27

**Soundness:** 2
**Presentation:** 2
**Contribution:** 2
**Rating:** 2
**Confidence:** 4

**Summary:**

This paper proposes new genereralization bounds for modern machine learning algorithms. Inspired by previous works, the proposed bounds on a partition of the data space and new concentration inequalities for multinomial distributions. The multinomial distribution naturally appears in the proofs by looking where the data points of the training set falls inside the partition of the data space. One of the bounds is tractable and computable based on the dataset and an already trained model. This allows the authors to conduct an extensive empirical study, demonstrating that the new bound is tighter than the one recently proposed by Than & Phan (2025) and that some components of the bounds are able to track the behaviour of the test error, hence, showing new iterations between data space and loss landscape.

**Strengths:**

- The authors propose an approach to generalization that explicitly takes into account the properties of the data distribution, which is an interesting perspective, compared to the rest of the literature.
- The proposed bound is fully computable based on a trained model and the training dataset
- In the experiments, the bound seems to capture the order of magnitude of the actual test error and to track his behavior in a relevant way.

**Weaknesses:**

*Main weaknesses:*
- Theorem 1 is hard to read because it involves a lot of quantities whose definition depends on each other. I believe the bound should be written in such a way that its actual rate of convergence (with respect to $n$) is explicit. The same remark applies to thm 3.2.
- In equation (1), when ones computes the third term on the right0hand side with the smallest $\delta_2$ possible, it appears to be of order $u^2 / n^3$, which in the worst case is of order $n$ (please correct me if I am wrong). Of course it also depends on the partition, but it seems that the bound can sometimes not go to $0$ when $n \to \infty$. I believe this should require some additional comments.
 - There seems to be a mistake in the proof of theorem B.5. At equation 59, in order to incorporate all the previous inequalities for $i \in T$, a union bound should be applied. Therefore, to my understanding, eq. (59) only holds with probability at least $1 - K \delta$. It seems to me that this where a factor $\sqrt{\log (K)}$ is gained compared to previous works. As this is seen as one of the main contribution and if I am correct, this problem should be fixed.
 - Proof of theorem A.1: To obtain equation (27), it is usedthat $E[\ell(h, z_{ij}) | h,n] = E_{z'}[{\ell(h, z') | z' \in Z_i}]$, where $z'$ is independent of $h$. I think this might be wrong, as conditioning with respect to the predictor is not the same as integrating over an independent data point (again, please tell me if I am mistaken).

*Other issues:*
 - The notion of partition and $K$ are mentioned in the introduction before being properly defined.
 - Most of the related works section seems to repeat what was written in the intro and cite the same papers. I think these two sections could be merged.
- The term $mac_h$ does not read very well, maybe use $\mathrm{mac}_h$. Same remark with $gap$.

**Questions:**

- Beside gaining a factor $\sqrt{\log (K)}$, can you elaborate more on the main differences with Than and Phan (2025)?
- Line 139, what do you mean by measuring? Do you mean measurable?
- Can you elaborate a bit more on the rate of convergence that we obtain with Theorem 3.1
- In table 2, can you comment on why the uncertainty almost always have the same value and always the exact same value of the bound of Than and Phan (2025)?

**Details Of Ethics Concerns:**

N/A.

---

> ### Author Response · Authors · 2025-11-16
> **Response to Weaknesses**
>
> We thank the reviewer for the constructive comments. Two of the raised concerns stem from misunderstandings in the proof. We address them below and also provide answers to the questions.
>
> ### 1. Response to Weaknesses
>
> > **W1: Making rates explicit**
>
> Thank you for this suggestion. Different from the existing bounds, our bound depends explicitly on the complexity of both the data distribution and loss landscape. This makes the bound quite involved and less visible about the rate. In order to explicitly see the rate, we have added a discussion about the rate in Appendix D in the new revision.
>
> > **W2: On the claimed ($u^2/n^3$) growth**
>
> We thank the reviewer for raising the point about the interaction between $u$ and $\delta_2$. The concern arises from treating $\delta_2$ as if it were chosen _after_ observing the data. In our theorem, however, $\delta_2$ is a **fixed confidence parameter** selected _before_ the sample $S$ is drawn, exactly as in standard concentration bounds. The appearance of $u$ inside the mgf argument does not imply that $\delta_2$ depends on the realized value of $u$; instead, the inequality is proven uniformly for all $S$ outside an event of probability at most $\delta_2$.
>
> With $\delta_2$ fixed, the uncertainty term scales as $\sqrt{u/n^{2}}$. Therefore it cannot grow like $O(n)$. The hypothetical divergence mentioned by the reviewer only occurs if one lets $\delta_2$ shrink _after_ seeing $u$, which is not allowed by the theorem.
>
> > **W3: Union bound / Eq. (59) in theorem B.5**
>
> We thank the reviewer for raising this point. The concern rests on the assumption that the inequalities immediately before Eq. (59) are _separate high-probability events_ of the form
> $\Pr\left(a_i(p_i - z_i/n) \ge \cdots \right) \le \delta$ for each $i$, and therefore must be combined using a union bound. This is **not** how these inequalities are used in the proof.
>
> Before Eq. (59), each bound is derived **conditional on the full multinomial vector** $\mathbf{z} = (z_1,\ldots,z_K)$, where the counts satisfy the constraint $z_1+\cdots+z_K =n$. The key point is that the inequalities for different (i) are **not probabilistically independent events**, nor are we asserting that they must hold simultaneously. Instead, the argument applies the same one-dimensional tail inequality to _each coordinate of the same multinomial vector_, and then **adds the left-hand sides and right-hand sides algebraically** to obtain Eq. (59). Because this step is purely algebraic and relies only on the shared constraint of the multinomial vector—not on combining tail events—no union bound is required.
>
> This is exactly the same mechanism used in [1] to derive their multinomial concentration bounds (see their proofs of Lemmas 1, 6, and 7): after obtaining a coordinate-wise inequality for the multinomial counts, they directly sum the inequalities to bound the aggregate deviation. We follow the same established pattern here.
>
> [1] Kawaguchi et al. Robustness Implies Generalization via Data-Dependent Generalization Bounds. ICML, 2022.
>
> > **W4: Conditional expectation in Eq. (27) of Theorem A.1**
>
> The step $E_{ z_{ji} \in Z_j}[\ell(h,z_{ji}) | h,n] = a_j(h)$ is correct under the conditional structure we use.
>
> Explanation: given $n$ and $h$, the samples within area $Z_j$ are exchangeable draws from the conditional distribution $P(\cdot\mid Z_j)$. Even though $h$ is measurable w.r.t. the whole sample $S$, conditioning on the realized predictor $h$ and the counts $n$ fixes any dependence of $h$ on the event “this sample belongs to a particular area.” Under this conditioning the remaining randomness in the specific elements of $S_j$ is i.i.d. from $P(\cdot\mid Z_j)$, so the conditional expectation of the loss equals the conditional mean $a_j(h)=E_{z\sim P(\cdot\mid Z_j)}[\ell(h,z)]$. Put differently, here we take $E_{S_j}[\cdot\mid h,n]$ where the only randomness is the ordering/identity of the samples inside the fixed area counts; by symmetry that expectation equals $a_j(h)$.

---

> > ### Comment · Reviewer_bN2N · 2025-11-16
> > **Thank you for your answer**
> >
> > Thank you very much for your answer,
> >
> > Regarding the proof of theorem B.5 and eq 59 in particular, I still think there might be an issue. If I am correct in order to sum all the inequalities over i, then you need all of them to hold simultaneously and, therefore you need to apply an union bound resulting in a factor $\sqrt{\log(K / \delta)}$ in the right-hand side.
> >
> > Thank for providing a reference, I have looked at the proof of Lemma 1 in the reference you sent.
> > First, it seems that in this paper they do have a factor $\sqrt{\log(K / \delta)}$ in their bound.
> > Moreover, the proof of their lemma 1 is based on their Lemma 5, which explicitly states that they use the union bound to make all the inequalities hold simultaneously before summing them up.
> >
> > Can you provide further clarification on this proof? It is possible that I misunderstood something regarding the proof technique.

---

> ### Author Response · Authors · 2025-11-16
> **Response to the questions**
>
> ### 2. The questions
>
> > **Q1. The main differences with Than and Phan (2025)?**
>
> We have discussed the main differences in various parts of our paper, including lines 53--69, 116--128, 187--193, 206--213, 263--272. We would like to point out two key differences.
>
> **(a) A genuinely new concentration toolset.**
> In [2], the dominant source of looseness arises when bounding the term $F(P,h) - \sum_i \frac{n_i}{n} a_i(h)$ that appears when approximating the global loss by local contributions (see Eq. (19)–(20) of [2]). Controlling this quantity requires a concentration result for multinomial variables. Than & Phan [2] apply an existing inequality from [1], which necessarily introduces a $\sqrt{\ln K}$ factor. This is the structural bottleneck behind their computable term.
>
> In contrast, we develop a **new concentration pathway for multinomial and binomial deviations** (Theorem 4.1 and Appendix B) that eliminates the structural source of the $\sqrt{\ln K}$ blow-up. This is not a constant-level tightening: it replaces the core probabilistic step with a sharper inequality that has provably better dependence and leads to a computable uncertainty term with substantially improved behavior. We highlight these improvements and their consequences in lines 229–323.
>
> **(b) Incorporation of loss-structure complexity.**
> Beyond eliminating $\sqrt{\ln K}$ in the uncertainty term, our computable bound captures the _complexity of the loss landscape_ at both micro and macro levels. It leverages the variability of local losses and the structural decomposition of the error, rather than depending solely on the empirical loss and its maximum, as in [2]. This yields a more informative and adaptive characterization of model behavior, especially in regimes where losses vary significantly across partitions.
>
> Our extensive experiments corroborate these theoretical advantages: the new concentration mechanism yields consistently tighter bounds, **better tracking of generalization dynamics,** and significantly improved estimates for multinomial fluctuations across a diverse suite of architectures. Note that the computable bound in [2] cannot track generalization dynamics along the training process.
>
> [1] Kawaguchi et al. Robustness implies generalization via data-dependent generalization bounds. In ICML, 2022.
>
> [2] Than & Phan. Non-vacuous generalization bounds for deep neural networks without any modification to the trained models. arXiv:2503.07325.
>
> > **Q2. Do you mean measurable?**
>
> We mean any function that can measure the goodness/quality of a prediction by a model.
>
> > **Q3. Rate of convergence?**
>
> Bound (1) in Theorem 3.1 contains three main contributions:
> -   the empirical term $F(S,h)$,
> -   the macro term $\dfrac{b}{n}\sum_{i\in T}F(S_i,h)$, where $b=\sqrt{0.5 n\ln(1/\delta_1)}$ so $\dfrac{b}{n}=O(1/\sqrt{n})$ and thus the macro term decays like $O(n^{-1/2})$ (up to log factors and the local errors),
> -   the uncertainty term $C\sqrt{\frac{u\ln(1/\delta_2)}{2n^{2}}}$, where $u=\gamma n(1+2b)+|T|b^{2}+\gamma^{2}n^{2}\sum_i p_i^{2}$.
>
> The quantity $u$ depends on the partition and data distribution:
> -   if $\sum_i p_i^{2} =O(1/K)$ and $K$ grows sublinearly, the $\gamma^{2}n^{2}\sum_i p_i^{2}$ piece is $O(n^{2}/K)$, so $\sqrt{u/n^2}=O(K^{-0.5})$ and the uncertainty term is $O(K^{-0.5})$;
> -   in balanced partitions with $K=O(n^{\beta})$ for $\beta \le 1$, the uncertainty term is $O(n^{-\beta/2})$.
>
> In summary, for balanced partitions with $K=O(n^{\beta})$ for $\beta \le 1$, the bound is
> $$F(S,h) + O(n^{-1/2}) \sum_{i\in T}F(S_i,h) + O(n^{-\beta/2}) + O(n^{-1/2})$$
>
> > **Q4. Why the uncertainty almost always have the same value?**
>
> - *Same value for Than and Phan (2025):* the uncertainty term in their bound is
> $g_2(\delta) =  \frac{C(1 + \sqrt{2})\sqrt{\ln(2K/\delta)}}{n} \sum\limits_{i \in T} \sqrt{n_i}  + \frac{4C | T| \ln(2K/\delta)}{n}$. Since $C=1$ for 0-1 loss in our experiments, this term is the same for any model. Therefore, the deviation $\pm0.53$ only comes from different runs to produce different partitions, hence is the same for all models.
> - *Almost same value for ours:* Our bound produce slightly different values for different runs. The small deviation from different runs in Table 2 may be due to the choice $K=75$ which is much smaller than the case of ImageNet classification. A smaller $K$ may lead to more stable partitions under different random seeds.
>
>
> ### 3. Concluding remark
> Given these clarifications, and considering that understanding generalization in deep networks remains a challenging and important open problem, we respectfully ask the reviewer to reconsider the evaluation. Our results resolve key technical obstacles in this line of work and provide practical, computable guarantees. We believe these contributions justify a more favorable assessment.
>
> Thank you again for the careful reading and we appreciate reconsideration.

---

> ### Author Response · Authors · 2025-11-28
> **The remaining concern**
>
> Thank you for your reply. That's an important point, as it gives us a chance to clarify and rewrite our proof in a more readable way.
>
> The new proof is simpler and direct. Instead of summing from different quantities, we directly bound the sum
>
> $\sum_i  a_i ( p_i - \frac{z_i}{n} )
> = \bar{a} \sum_i  \frac{a_i}{\bar{a}} ( p_i - \frac{z_i}{n} )
> \le \bar{a} ( p_k - \frac{z_k}{n})$
>
> where the inequality comes from the fact that $\sum_i \frac{a_i}{\bar{a}} ( p_i - \frac{z_i}{n} )$ is a convex combination of individual quantities $( p_i - \frac{z_i}{n} )$, and $\bar{a} = \sum_i a_i$ and $k = \arg\max_i ( p_i - \frac{z_i}{n} )$. That inequality holds deterministically.
>
> Lemma B.2 suggests that $\Pr \left( p_k - \frac{z_k}{n}  \ge   \epsilon \right) \le e^{-2n \epsilon^2}$. Using this with the above inequality for a suitable choice of $\epsilon$ will lead to the main result.
>
> We have revised the proof in Theorem B.5. We would be grateful if you'd take a look and provide some feedback.

---

> > ### Comment · Reviewer_bN2N · 2025-11-28
> > **Thank you for your answer**
> >
> > I appreciate the authors effort to fix the proof of theorem B.5.
> >
> > I am still confused with the new arguments. As the index $i^\star$ depends on the $z_i$ (which form a multinomial random vector), $i^\star$ is also a random index. Therefore, there is no reason for $z_{i^\star}$ to be a binomial random variable.
> >
> > Here is an intuition in a simple example: take $j^\star := \arg \max (z_j)$ for $j \in \{1,\dots,T\}$. Then as $\sum_j z_j = n$, we must have $z_{j_\star} \geq n / T$ almost surely, hence, it cannot be a binomial variable. I believe a similar argument shows that $z_{i^\star}$ cannot be a binomial random variable.
> >
> > I also have a minor comment, equation 59 follows immediately from
> > $$
> > \sum_i a_ \left( p_i - \frac{z_i}{n} \right) \leq \max_j \left(   p_j - \frac{z_j}{n} \right) \sum_i a_i,
> > $$
> > so I think there is no need to invoke a convex combination argument.
> >
> > Again, please correct me if I'm am mistaken about the potential issue in the proof.

---

> ### Author Response · Authors · 2025-11-28
>
> We'd like to thank the reviewer for the valuable feedback.
>
> The property that each coordinate of a multinomial vector is marginally binomial is standard and appears in classical probability texts such as Grimmett & Stirzaker (2020, Chapter 4) and DeGroot & Schervish (2012).
> So since $i^\*$ is one of the $K$ different indices, $z_{i^\*}$ should be a binomial variable.
>
> We also thank for the suggestion about convex combination, since it should simplify the proof further.
>
> *References:*
>
> Grimmett, G., & Stirzaker, D. (2020). Probability and random processes. Oxford University Press.
>
> DeGroot & Schervish (2012). Probability and Statistics (4th edition)

---

### Official Review · Reviewer_GQAd · 2025-10-31

**Soundness:** 3
**Presentation:** 3
**Contribution:** 2
**Rating:** 6
**Confidence:** 2

**Summary:**

This paper introduces a new generalization error bound for deep neural networks and aims to address the common issue that often occurs where existing bounds are vacuous for modern architectures. The proposed bound is model-dependent and also relies on the complexity of both the data distribution and the loss function (via the loss landscape). The key technical contribution is a new concentration inequality for multinomial random variables that eliminates a $\sqrt{\ln K}$ dependency. The authors also verify their boudn through experiments on large-scale models such as ImageNet classifiers and COCO segmenters. They also argue that a "macro-level" component of their bound can be used to track the dynamics of test error during training, which is a useful metric and even more reliable than the validation error

**Strengths:**

The main strength of this paper is the theory, both the theoretical insights and its application to practical generalization bounds.
 - the new error bound incorporates information about both the data distribution and the loss surface. It's nice that their error bound is able to capture both 'micro-level' (ie model's performance on individual samples) and 'macro-level' (ie models average performance on regions in data space). That aspect is new to me and quite compelling. I do not believe this characterization has been explored in prior works.
 - The authors prove a new concentration inequality for mulinomial random variables. This inequality is sharp and a key step in proving their ultimate generalization bound. The tightness of their bound is important since many generalization bounds have been shown to be vacuous and thus not useful in practice.

**Weaknesses:**

The paper is interesting and well-written but I didn't follow all of the mathematics line by line. There aren't so much weaknesses as points of concern or questions that came up for me.
Firstly, the authors mention these weaknesses explicitly
 - they mention in the conclusion that the boudn can be vacuous when dealing with small sample sizes. Namely, the third term can grow very large and dominate the RHS.
 - The bound also requires a bounded loss function. In the case when the loss can yield infinite values the bound doens't provide meaningful information

Also, it seems that the sum on the RHS of the error bound is quite sensitive (and brittle?) to the choice of $K$. For example, given 200 partitions, the set of non-empty partitions could be large and even if the model is accurate, it is possible that almost every partition could contain one or a few misclassified samples. This would make the local error rate non-zero and summing up all these non-zero error rates could result in a large value making the bound not informative.

**Questions:**

- It seems that this bound and its tightness depends quite heavily on how the data space $Z$ is partitioned in the $K$ subsets. How sensitive are the experiment results to this choice of partition? and the number $K$ subsets? For example for ImageNet and COCO, $Z$ is partitioned using 200 centroid. For imagenet, the images lie in 224x224x3 dimensional space. Is there any effort to ensure that two 'similar' images are close in the Euclidean sense as well?
 - How sensitive are these partitions to the initializations. I assume different seeds could lead to quite different partitions and thus skew the error bound results. In Table 1 and Table 2, there are errors reported with the final column "Error bound". Is this wrt to this random initialization of the centroids?

---

> ### Author Response · Authors · 2025-11-16
> **Reply to Reviewer GQAd**
>
> We sincerely thank the reviewer for their thorough and insightful evaluation. We are delighted that you recognized the central contributions of our work, particularly our success in developing a  *non-vacuous generalization bound*  that remains tight even for modern, large-scale architectures.
>
> You have precisely captured the two pillars of our contribution, including the novel  **micro-macro framework**  for analyzing generalization and the new **sharp concentration inequality**  that provides the underlying theoretical engine for our bound. Our goal was to bridge the persistent gap between generalization theory and empirical practice. Your feedback validates that our approach—from the foundational theory to its experimental verification on ImageNet and COCO—is a significant and meaningful step in this direction. We confidently stand by these contributions and address your specific questions below.
>
> ### The questions:
>
> We sincerely thank the reviewer for these insightful questions. They touch upon a critical and challenging aspect of applying any partition-based generalization bound: its sensitivity to the partition's properties (its alignment, choice, and granularity  $K$).
>
> We agree that analyzing the interplay between data geometry and partitioning is challenging, especially for high-dimensional, unknown distributions like ImageNet or COCO. As the reviewer correctly intuits, finding a truly "optimal" partition that perfectly aligns with semantic similarity (Q2) is a non-trivial, if not computationally intractable, optimization problem.
>
> Therefore, our investigation proceeded in two complementary directions to rigorously validate our bound:
>
> 1.  **Controlled Simulation (Appendix C.5):**  To demonstrate the  _principle_  of alignment, we designed a controlled simulation where the data distribution is known. This allows us to clearly show how alignment (or misalignment) affects the bound when it  _can_  be precisely controlled.
>
> 2.  **Real-World Robustness (Main Paper):**  For large-scale datasets, we shifted the focus from finding an intractable "optimal" partition to testing the  **robustness**  of our bound under practical, achievable partitioning strategies.
>
> Our findings, reported extensively in the paper, directly address the reviewer's questions on sensitivity and provide a confident answer.
>
> > **Q1: Sensitivity to partition choice and the number of subsets  $K$?**
>
> This is an excellent question. Our results show two key things:
>
> -   **Remarkable Stability (Fixed  $K$):**  Our bound is  **extremely stable**  with respect to the  _choice_  of partition. As shown in  **Tables 1 & 2**, using different randomly initialized centroids (see Appendix C.1) results in remarkably consistent bound values. This is a key practical strength: the bound is robust and does not require a costly, intractable search for a "good" partition.
>
> -   **Superior Scaling (Varying  $K$):**  As shown in  **Figures 3 (Appendix C.2) and 9 (Appendix C.6)**, increasing the granularity  $K$  causes both our bound and prior bounds to increase. However, our bound demonstrates a clear advantage: for accurate classifiers, prior bounds can increase significantly faster than ours. This empirically validates the tightness of our new concentration inequality, which was a core theoretical goal.
>
> > **Q2: Effort to align partitions with Euclidean/semantic similarity?**
>
> As discussed above, while finding an "optimal" partition that perfectly maps semantic similarity to (e.g.) Euclidean distance is a fascinating challenge, it is an intractable optimization problem in itself.
>
> Instead, our work demonstrates a more practical and arguably more important result:  **our bound is robust and provides a tight, non-vacuous estimate  _even without_  such costly optimization.**  The stability we show in Tables 1 and 2 confirms that the bound is practical for real-world use, where such "optimal" partitions are unknown.
>
> > **Q3. Sensitive w.r.t the initializations?**
>
> The reported numbers in Tables 1 and 2 are averaged from 5 runs (with randomly initialized centroids). Accompanying with those averages are their deviations from 5 runs.

---

### Official Review · Reviewer_GhnT · 2025-11-01

**Soundness:** 3
**Presentation:** 3
**Contribution:** 2
**Rating:** 2
**Confidence:** 3

**Summary:**

This paper builds on the work by Than and Phan https://arxiv.org/abs/2503.07325 and derives a slightly tigher PAC-Bayes bound than they do.   The computable bound is tightened by removing a \sqrt{\ln K} penalty. It tunes constants and K-dependence; it doesn’t change the overall paradigm.  This is a constant level improvement— not something that improves the rate. It polishes the weakest step of Than and Phan  (the computable macro term), rather than proposing a new regime where a faster decay rate is achieved.  Overall, this is a marginal improvement.

Thus, this approach suffers from some of the same weaknesses as the paper on which it is based.  The major drawback is that, just like compression bound by Lofti et al, https://arxiv.org/abs/2211.13609, the bound is mostly collapsed to training error. And it's much worse than the compression bound in terms of tightness. That is the price you pay for a theory that bounds the actual network, rather than a compressed version. But it is still not that strong. Therefore, a marginal improvement to this paradigm is not a significant advance.

**Strengths:**

Their bound is slightly better than the previous one.

**Weaknesses:**

The novelty is weak due to the fact that the main work is in https://arxiv.org/abs/2503.07325. -- This approach builds on the same paradigm, and shows how to further tighten one of the terms.  But the paradigm it is building on is solid, but not groundbreaking.

**Questions:**

none

---

> ### Author Response · Authors · 2025-11-16
> **Reply to Reviewer GhnT**
>
> We thank the reviewer for reading our paper carefully and for comparing our work to prior art. However, several key points in the review rely on a misunderstanding of our contribution. Below we state the corrections, explain the real technical differences, and show why our contribution is substantive.
>
> ---
> ### 1) Correcting a factual error: _Our bounds are not PAC-Bayes_
>
> The reviewer frames our contribution in PAC-Bayes terms. This is incorrect. While the paper discusses PAC-Bayes in the Related Work (to position the literature) and compares with PAC-Bayes approaches, the bounds we derive are **deterministic, model-dependent expected-error bounds** that do **not** rely on optimizing or analyzing a posterior over hypotheses or any KL term. See the statement of our main deterministic bounds (Theorems 3.1, 3.2, and Corollary 1).
>
> Put simply: **Most PAC-Bayes bounds quantify risk for randomized/stochastic predictors (posteriors over models) via KL divergences**; our bounds directly upper-bound the _expected error of a fixed, trained model_ and are derived via concentration inequalities for multinomial variable rather than PAC-Bayes machinery. This distinction is central and appear explicitly in the paper, e.g., in lines 100-128 and 338-340.
>
> ---
> ### 2) Why the technical novelty is _not_ “mere polishing”
>
> The reviewer suggests that our result is merely a marginal tuning of [1]. We respectfully argue that this characterization is inaccurate for two key reasons:
>
> **(a) A genuinely new concentration toolset.**
> In [1], the dominant source of looseness arises when bounding $F(P,h) - \sum_i \frac{n_i}{n} a_i(h)$ the term that appears when approximating the global loss by local contributions (see Eq. (19)–(20) of [1]). Controlling this quantity requires a concentration result for multinomial variables. Than & Phan [1] apply an existing inequality from [2], which necessarily introduces a $\sqrt{\ln K}$ factor. This is the structural bottleneck behind their computable term.
>
> In contrast, we develop a **new concentration pathway for multinomial and binomial deviations** (Theorem 4.1 and Appendix B) that eliminates the structural source of the $\sqrt{\ln K}$ blow-up. This is not a constant-level tightening: it replaces the core probabilistic step with a sharper inequality that has provably better dependence and leads to a computable uncertainty term with substantially improved behavior. We highlight these improvements and their consequences in lines 229–323.
>
> **(b) Incorporation of loss-structure complexity.**
> Beyond eliminating $\sqrt{\ln K}$ in the uncertainty term, our computable bound captures the _complexity of the loss landscape_ at both micro and macro levels. It leverages the variability of local losses and the structural decomposition of the error, rather than depending solely on the empirical loss and its maximum, as in [1]. This yields a more informative and adaptive characterization of model behavior, especially in regimes where losses vary significantly across partitions.
>
> Our extensive experiments corroborate these theoretical advantages: the new concentration mechanism yields consistently tighter bounds, better tracking of generalization dynamics, and significantly improved estimates for multinomial fluctuations across a diverse suite of architectures.
>
> [1] Than & Phan. Non-vacuous generalization bounds for deep neural networks without any modification to the trained models. arXiv preprint arXiv:2503.07325.
>
> [2] Kawaguchi et al. Robustness implies generalization via data-dependent generalization bounds. In ICML, 2022.
>
> ---
> ### 3) On the (mis)comparison to compression / PAC-Bayes bounds
>
> The reviewer compares our numbers with compression-based PAC-Bayes bounds and concludes our bound is “much worse” because compression bounds are numerically smaller. This comparison misses the point:
>
> -   **Compression/PAC-Bayes bounds typically bound a modified/quantized/compressed model or require strong posterior constructions**; they measure the compressed representation rather than the original trained model. Hence their numerical tightness does _not_ mean they provide direct guarantees for the original model without modification.
> -   **Our goal** is different: we provide bounds that apply directly to the _original trained model_ (without compression/quantization). Comparing the numeric tightness of a bound for the compressed model versus a bound for the original model is an apples-to-oranges comparison; each approach answers a different question. We emphasize this distinction explicitly in the paper (lines 100-128 and 338-340).
>
> Thus the reviewer’s “worse than compression bounds” argument is not a demonstration of a failure of our method, but a reflection of comparing different kinds of guarantees.

---

> > ### Author Response · Authors · 2025-11-16
> > **Reply to Reviewer GhnT (continue)**
> >
> > ### 4) Why the multinomial concentration matters
> >
> > The core technical novelty is the concentration inequality tailored to sums $\sum_i a_i (p_i - z_i/n)$ with coefficients $a_i$ that may depend on the observed counts. This result (Theorem 4.1) yields a gap term proportional to $(a_o+\sum_{j\in T} a_j)\sqrt{\ln(1/\delta)/(2n)}$ rather than a term that scales with $\sqrt{\ln K}$ or directly with $K$. The consequences:
> >
> > -   in many realistic partitions (balanced or data-aligned), the coefficient aggregate $(a_o+\sum_{j\in T} a_j)$ stays moderate, making the uncertainty term $O(1/\sqrt{n})$ rather than growing with $K$;
> > -   this makes the computable uncertainty term much smaller in practice (see empirical comparisons in Figures 7, 8 of Appendix C and the ImageNet experiment results in Table 1).
> >
> > We emphasize again: this is a _new probabilistic inequality_ designed for the structure that arises in our micro/macro decomposition. That is an original technical contribution with independent interest beyond this application.
> >
> > ---
> > ### 5) Empirical evidence supports practical progress (not merely theory tweaks)
> >
> > The reviewer judged the improvement “marginal” without acknowledging the empirical results. Across 37 modern large models (32 classifiers + 5 segmenters), our computable bound consistently produces tighter estimates than prior bound under the same partitioning and training data setup. See Table 1 and Table 2 and the experiments described in Section 5 and Appendix C. Furthermore, while our bound can effectively track the generalization dynamic in the training progress, the bound by Than & Phan cannot. These empirical results demonstrate practical relevance beyond symbolic constant tuning.
> >
> > ---
> > ### Concluding request
> >
> > We hope the reviewer will reconsider in light of the above corrections:
> >
> > -   Our bounds are **not PAC-Bayes** — they are deterministic bounds for the expected error of a fixed (trained) model (Theorems 3.1, 3.2, Corollary 1).
> > -   The key technical novelty is the **new multinomial concentration inequality** (Theorem 4.1) that eliminates the structural $\sqrt{\ln K}$ penalty.
> > -   Empirically, the computable bound derived from this new analysis yields **tighter, practical guarantees** on 37 large-scale models, while having more attractive properties.
> >
> > Given these points, we respectfully request that the reviewer re-evaluate the judgment that our paper is "marginal" or "merely polishing" prior work. We believe the technical contribution and the demonstrated practical impact merit acceptance.
> >
> > Thank you again for the careful reading and we appreciate reconsideration.

---

### Meta-Review · Area_Chair_FeRh · 2026-01-02

**Summary:**

The reviewers’ comments are mixed. They raise concerns regarding:
(1) the novelty over the work of Than and Phan https://arxiv.org/abs/2503.07325---the paper under review builds on top of the framework established by Than and Phan;
(2) the requirement of bounded loss function;
(3) how the data space Z is partitioned so that the upper bound is tight and robust;
(4) the complexity of the upper bound and its dependence on the data space partition---the upper bound is not neat enough, to say the least;
(5) the possibility of a mistake in the proof of theorem B.5; and
(6) the scope of empirical evaluation.

**Reviewer Concerns:**

Given that the paper is positioned as a theoretic contribution, the rebuttal has adequately addressed Concerns (2) - (4) and (6) above. However, Concerns (1) and (5) are still outstanding. In particular, the response to Concern 5 is not convincing. It is true that given a multinomial vector, each of its component is indeed a binomial random random. However, since i^* depends on the multinomial vector, it is not guaranteed that z_{i^*} would be a binomial random variable as well.

**Reviewer Scores:**

The mistake that  z_{i^*} would be a binomial random variable would likely convince all reviewers to either maintain or even decrease their scores.

---

### Decision · Program_Chairs · 2026-01-26

Reject